# DRDT3: Diffusion-Refined Decision Test-Time Training Model

**Xingshuai Huang**  *xingshuai.huang@mail.mcgill.ca*
*Department of Electrical and Computer Engineering*
*McGill University*

**Di Wu**  *di.wu5@mcgill.ca*
*Department of Electrical and Computer Engineering*
*McGill University*

**Benoit Boulet**  *benoit.boulet@mcgill.ca*
*Department of Electrical and Computer Engineering*
*McGill University*

**Reviewed on OpenReview:** *https://openreview.net/forum?id=I6zjLhIzgh*

## Abstract

Decision Transformer (DT), a trajectory modelling method, has shown competitive performance compared to traditional offline reinforcement learning (RL) approaches on various classic control tasks. However, it struggles to learn optimal policies from suboptimal, reward-labelled trajectories. In this study, we explore the use of conditional generative modelling to facilitate trajectory stitching given its high-quality data generation ability. Additionally, recent advancements in Recurrent Neural Networks (RNNs) have shown their linear complexity and competitive sequence modelling performance over Transformers. We leverage the Test-Time Training (TTT) layer, an RNN that updates hidden states during testing, to model trajectories in the form of DT. We introduce a unified framework, called **D**iffusion-**R**efined **D**ecision **TTT** (DRDT3), to achieve performance beyond DT models. Specifically, we propose the Decision TTT (DT3) module, which harnesses the sequence modelling strengths of both self-attention and the TTT layer to capture recent contextual information and make coarse action predictions. DRDT3 iteratively refines the coarse action predictions through the generative diffusion model, progressively moving closer to the optimal actions. We further integrate DT3 with the diffusion model using a unified optimization objective. With experiments on multiple tasks in the D4RL benchmark, our DT3 model without diffusion refinement demonstrates improved performance over standard DT, while DRDT3 further achieves superior results compared to state-of-the-art DT-based and offline RL methods.

## 1 Introduction

Reinforcement learning aims to train agents through interaction with the environment and receiving reward feedback. It has been widely applied in areas such as robotics (Jia et al., 2023; Guo et al., 2023), autonomous driving (Liu et al., 2024a), energy systems (Zhang et al., 2022; Bui et al., 2025), transportation systems (Zhai et al., 2025), and games (Barros e Sá & Madeira, 2025). Offline RL, also known as batch RL, eliminates the reliance on online interactions with the environment (Wang et al., 2022) by directly learning policies from offline datasets composed of historical trajectories. Therefore, it is promising for applications where interactions are risky and costly. Directly applying dynamic programming-based online RL methods to offline RL problems typically learns a poor-quality policy due to overestimated out-of-distribution actions caused by distribution shift (Yamagata et al., 2023). Conventional offline RL generally addresses such issues

by regularizing policy to stay close to behavior policy (Lyu et al., 2022) or constraining values of out-of-distribution actions (Kostrikov et al., 2021; Liu et al., 2024b).

DT (Chen et al., 2021), an innovative offline RL approach that formulates policy learning as a sequence modelling problem, has shown superior performance over conventional offline RL methods on some benchmarks. By leveraging the strong sequence modelling capacity of Transformer structures (Vaswani et al., 2017; Radford et al., 2019), DT models trajectories autoregressively, bypassing the necessity for bootstrapping and learning a value function for a given state. Fed with the recent context, a historical subtrajectory composed of states, actions, and return-to-go (RTG) (cumulative rewards from the current step), DT predicts actions autoregressively. However, the abandonment of dynamic programming in DT makes it lack stitching ability (Yamagata et al., 2023). Some recent work studies this problem. For instance, Q-learning DT (Yamagata et al., 2023) relabels RTG with a value function learned from Q-learning, enhancing DT to learn better policies from sub-optimal datasets. Elastic DT (Wu et al., 2023) adjusts the context length according to the evaluated optimality of the context to achieve trajectory stitching. Our work instead addresses such a problem using generative modelling.

Recently, diffusion models (Ho et al., 2020; Cao et al., 2024; Yan et al., 2024) have gained recognition for their extraordinary ability to generate high-quality complex data, such as images and texts. Additionally, recent works have also introduced the diffusion model to offline RL (Janner et al., 2022; Ajay et al., 2023; Ho et al., 2020). Similar to DT, Diffuser (Janner et al., 2022) also works as a trajectory generator, using expressive diffusion models instead of Transformers. Unlike existing diffusion-based offline RL methods, our work leverages diffusion models as a refinement tool.

We propose DRDT3, a framework that combines a novel DT-style trajectory modelling method and a conditional diffusion model. Specifically, we first introduce a DT3 module, which harnesses both self-attention and the TTT layer to achieve reduced complexity and improved sequence modelling ability. This module predicts coarse action representations based on recent context, which are then used as prior knowledge or conditions for a denoising diffusion probabilistic model (DDPM) (Ho et al., 2020). Additionally, we present a novel gated MLP noise approximator, designed to capture essential information from both noisy input and coarse action predictions, facilitating the denoising process. To integrate the DT3 module and diffusion model effectively, we employ a unified optimization objective with two key components: an action representation term, which ensures that the DT3 module generates coarse action conditions close to the optimal action distribution, and an action refinement term that constrains the diffusion model to sample actions within the dataset distribution. This approach enables the predicted actions from DT3 to be iteratively refined with Gaussian noise through the denoising chain within the diffusion model, facilitating trajectory stitching.

The contributions of this work are summarized as follows:

**(i)** We introduce the DT3 model, which combines the self-attention mechanism and the TTT layer for improved action generation and enhanced performance over DT.

**(i)** We further propose DRDT3, a diffusion-refined DT3 algorithm that leverages coarse action representations from the DT3 module as priors and uses diffusion models to iteratively refine these actions. Additionally, we present a gated MLP noise approximator for more effective denoising. DRDT3 integrates the DT3 and diffusion model into a cohesive framework with a unified optimization objective.

**(iii)** Experiments on extensive tasks from the D4RL benchmark (Fu et al., 2020) demonstrate the superior performance of our proposed DT3 and DRDT3 over conventional offline RL and DT-based methods.

The following sections are organized as follows: Section 2 provides an overview of recent related work on DT, sequence modeling, and diffusion models in RL. We then present preliminaries in Section 3, covering offline RL, DT, the TTT layer, and diffusion models. Our DRDT3 methodology is detailed in Section 4, followed by experiments evaluating the proposed models in Section 5. Finally, we present conclusions and discussions in Section 6.

## 2 Related Work

### 2.1 Decision Transformer

Offline RL learns policies from historical trajectories pre-collected using other behavior policies. Different from classic offline RL (Fujimoto & Gu, 2021; Kostrikov et al., 2021) where dynamics programing is employed for policy optimization, DT (Chen et al., 2021) formulates the policy learning as a sequence modelling problem by autoregressively generating trajectories with Generative Pre-trained Transformer (GPT)-like architectures (Radford et al., 2019). Many variants of DT have been studied and shown competitive performance on some RL benchmarks. Online DT (Zheng et al., 2022) equips DT with entropy regularizers, enabling online fine-tuning with efficient exploration. Prompting DT (Xu et al., 2022) and Hyper-DT (Xu et al., 2023) make DT easily transferable to novel tasks using a trajectory prompt and an adaptation module initialized with a hyper-network, respectively. Hierarchical DT (Correia & Alexandre, 2023) frees DT from specifying RTGs while using sub-goal selection instead. Q-learning DT (Yamagata et al., 2023) and Elastic DT (Wu et al., 2023) improve the stitching ability of DT by introducing a dynamic programming-derived value function and varying-length context, respectively. DT has also been applied to some real-world applications, such as robotics (Correia & Alexandre, 2023) and traffic signal control (Huang et al., 2023).

### 2.2 Sequence Modeling

Transformer-based methods leveraging the attention mechanism have demonstrated remarkable performance across various sequence modelling domains (Vaswani et al., 2017; Radford et al., 2019; Achiam et al., 2023; Peebles & Xie, 2023). However, their quadratic time complexity during inference (Gu & Dao, 2023) poses challenges for long-sequence modelling. Recently, structured state space models (SSMs) (Gu et al., 2021; Gu & Dao, 2023) have introduced a compelling architecture for long-sequence processing, distinguished by their linear computational complexity. Among SSM-based methods, MAMBA has proven effective and is widely applied in various research areas (Zhang et al., 2024; Wang et al., 2024). MAMBA (Gu & Dao, 2023) introduces a selection mechanism to adapt SSM parameters based on input, along with parallel scanning, kernel fusion, and recomputation techniques for efficient computation. Other recent sequence modelling approaches, such as RWKV (Peng et al., 2023; 2024), Gated Linear Attention (GLA) (Yang et al., 2023), and xLSTM (Beck et al., 2024), build on RNN architectures with enhanced expressiveness by utilizing matrix hidden states, unlike the vector hidden states of traditional RNNs. Additionally, TTT layers (Sun et al., 2024) further enhance hidden state expressiveness by updating them through self-supervised learning during both training and inference. TTT-Linear and TTT-MLP, whose hidden states are a linear layer and an MLP, respectively, achieve linear time complexity with performance that is comparable to, or even surpasses, Transformers and MAMBA.

### 2.3 Diffusion Models in Reinforcement Learning

Diffusion models (Ho et al., 2020) have been appealing for their strong capacity to generate high-dimensional image or text data. In light of the expressive representation and strong multi-modal distribution modelling ability of the diffusion model, some researchers have introduced the diffusion model to RL paradigms. Diffuser (Janner et al., 2022) first employs diffusion models as a planner for generating trajectories in model-based offline RL, which alleviates the severe compounding errors of conventional planners. AdaptDiffuser (Liang et al., 2023) improves the diffusion model-based planner by self-evolving with filtered high-quality synthetic demonstrations. In addition to employing diffusion models as planners, SynthER (Lu et al., 2023), DiffStich (Li et al., 2024), GODA Huang et al. (2024), and PRIDE Feng et al. (2025) further exploit diffusion models as data synthesizers to directly augment either offline or online training data with higher diversity and quality.

Diffusion models have also been adopted to represent policies. Decision Diffuser (Ajay et al., 2023) formulates sequential decision-making problems as conditional generative modeling and introduces constraints and skill conditions. Diffusion-QL (Wang et al., 2022) explores representing policy as a diffusion model and employing Q-value function guidance during training. It overcomes the over-conservatism of policies learned from conventional offline RL. Efficient Diffusion Policy (EDP) (Kang et al., 2023) extends diffusion models to policy gradient methods and accelerates training by approximating actions from corrupted samples in a single

step. Diffusion Actor-Critic (DAC) (Fang et al., 2024) adopts an actor-critic training framework, training a critic network alongside a diffusion-based KL-constrained policy, where soft Q-guidance is used to regularize the policy to remain close to the behavior policy. Ma et al. (2025) develop two diffusion-based online RL algorithms, i.e., Diffusion Policy Mirror Descent (DPMD) and Soft Diffusion Actor-Critic (SDAC), built on two novel reweighted score matching methods. MaxEntDP (Dong et al., 2025) and DIME (Celik et al., 2025) incorporate maximum entropy RL to enhance online exploration with diffusion-represented policies. DRCORL (Guo et al., 2025) applies diffusion-based policies to constrained RL and introduces a gradient operation to balance reward and safety constraints. FDPP (Chen et al., 2025) and PRIDE (Feng et al., 2025) further explore the application of diffusion models in preference-based RL. Moreover, CleanDiffuser (Dong et al., 2024) offers a diffusion model library that facilitates the development of diffusion-based decision-making methods.

Our DRDT3 uses a unified framework that aims to enhance a DT-style trajectory modeling method by refining the predictions iteratively within the expressive DDPM. In contrast to the above methods, DRDT3 employs the diffusion model solely as an action refinement module, directly operating on the DT output without multi-stage training. It remains a sequence-modeling approach without dynamic programming, avoiding the compounding errors of dynamics models and value estimation.

## 3 Preliminaries

### 3.1 Offline Reinforcement Learning and Decision Transformer

RL trains an agent by interacting with an environment that is commonly formulated as a Markov decision process (MDP): $M = \{\mathcal{S}, \mathcal{A}, \mathcal{R}, \mathcal{P}, \gamma\}$. The MDP consists of state $s \in \mathcal{S}$, action $a \in \mathcal{A}$, reward $r = \mathcal{R}(s, a)$, state transition $\mathcal{P}(s'|s, a)$, and discount factor $\gamma \in [0, 1)$ (Sutton & Barto, 2018). The objective of RL is to learn a policy $\pi$ that maximizes the expected cumulative discounted rewards:

$$J = \mathbb{E}_\pi \left[ \sum_{t=0}^{\infty} \gamma^t \mathcal{R}(s_t, a_t) \right], \tag{1}$$

where $s_t$ and $a_t$ denote state and action at time $t$, respectively. Offline RL typically learns a policy from offline datasets composed of historical trajectories collected by other behavior policies, eliminating the need for online interactions with the environment. A trajectory is a sequence of states, actions, and rewards up to time step $T$:

$$\tau = (s_1, a_1, r_1, ..., s_T, a_T, r_T). \tag{2}$$

Different from conventional dynamic programming-based offline RL methods, DT (Chen et al., 2021) formulates the policy as a return-condition sequence model represented by GPT-like architectures (Radford et al., 2019). Instead of generating the action based on a single given state, DT autoregressively predicts actions based on the recent context, a $K$-step sub-trajectory $\tau$ including historical tokens: states, actions, and RTGs $\hat{g}_t = \sum_{t'=t}^{T} r_{t'}$, where the RTG specifies the desired future return. The trajectory generation process can be formulated as

$$\tau' = (\hat{g}_{t-K+1}, s_{t-K+1}, a_{t-K+1}, \ldots, \hat{g}_t, s_t, a_t). \tag{3}$$

Specifically, at time step $t$, DT predicts $a_t$ based on $(\hat{g}_{t-K+1}, s_{t-K+1}, a_{t-K+1}, \ldots, \hat{g}_t, s_t)$ and repeats the procedure till the end.

### 3.2 Test-Time Training Layer

Due to the quadratic complexity of self-attention in processing long-sequence inputs, some recent work leverages the RNN structure for sequence modeling, given its linear complexity. However, RNNs are often constrained by their limited expressive power in hidden states. The TTT layer (Sun et al., 2024) aims to address this challenge by updating the hidden state on both training and testing sequences with self-supervised learning. Concretely, each hidden state can be expressed as $W_t$, the weights of a model $f$ that

produces the output $z_t = f(x_t; W_t)$. The update rule for $W_t$ is expressed as a step of gradient descent on a certain self-supervised loss $\ell$:

$$W_t = W_{t-1} - \eta \nabla \ell \left( W_{t-1}; x_t \right), \tag{4}$$

where $\eta$ is the learning rate and $x_t$ is the input token. TTT layer (Sun et al., 2024) sets $\ell$ to a reconstruction loss of $x_t$:

$$\ell \left( W; x_t \right) = \| f \left( \tilde{x}_t; W \right) - x_t \|^2, \tag{5}$$

where $\tilde{x}_t = \theta_K x_t$ denotes a low-rank projection corrupted input with a learnable matrix $\theta_K$. To selectively remember information in $x_t$, TTT reconstructs another low-rank projection $\theta_V x_t$ instead. Therefore, the self-supervised loss becomes:

$$\ell \left( W; x_t \right) = \| f \left( \theta_K x_t; W \right) - \theta_V x_t \|^2. \tag{6}$$

To make the dimensions consistent, the output rule becomes $z_t = f(\theta_Q x_t; W_t)$. A sequence model with TTT layers consists of two training loops: the outer loop for updating the larger network and hyperparameters $\theta_Q, \theta_K$ and $\theta_V$, and the inner loop for updating $W$ within each TTT layer. TTT further makes updating parallelizable through mini-batch TTT and accelerates computation using the dual form technique (Sun et al., 2024).

## 3.3 Diffusion Model

Diffusion models (Sohl-Dickstein et al., 2015; Ho et al., 2020) typically learns expressive representation of the data distribution $q(\mathbf{x}^0)$ from a dataset in the form $p_\theta \left( \mathbf{x}^0 \right) := \int p_\theta \left( \mathbf{x}^{0:N} \right) d\mathbf{x}^{1:N}$, where latent variables $\mathbf{x}^1, ..., \mathbf{x}^N$ from timestep $i = 1$ till $i = N$ have the same dimensionality as data $\mathbf{x}^0 \sim q(\mathbf{x}^0)$. Diffusion models typically consist of two processes: diffusion or forward process, and reverse process. The forward process is defined as a Markov chain, where the data $\mathbf{x}^0 \sim q(\mathbf{x}^0)$ is gradually added with Gaussian noise following a variance schedule $\beta_1, ..., \beta_N$:

$$q \left( \mathbf{x}^i \mid \mathbf{x}^{i-1} \right) := \mathcal{N} \left( \mathbf{x}^i; \sqrt{1 - \beta_i} \mathbf{x}^{i-1}, \beta_i \mathbf{I} \right). \tag{7}$$

The Gaussian transition can be alternatively formulated as $\mathbf{x}^i = \sqrt{\alpha^i} \mathbf{x}^{i-1} + \sqrt{1 - \alpha^i} \boldsymbol{\epsilon}^i$, where $\alpha^i = 1 - \beta^i$ and $\boldsymbol{\epsilon}^i \sim \mathcal{N}(\mathbf{0}, \mathbf{I})$. The simplified transition from $\mathbf{x}^0$ to $\mathbf{x}^i$ can be inferred that

$$\mathbf{x}^i = \sqrt{\bar{\alpha}^i} \mathbf{x}^0 + \sqrt{1 - \bar{\alpha}^i} \boldsymbol{\epsilon}(\mathbf{x}^i, i), \tag{8}$$

where $\bar{\alpha}^i = \prod_{i=1}^{i} \alpha^i$ and $\boldsymbol{\epsilon}(\mathbf{x}^i, i) \sim \mathcal{N}(\mathbf{0}, \mathbf{I})$ denotes the cumulative noise accumulated over timesteps. The reverse process denoises a starting noise sampled from $p \left( \mathbf{x}^N \right) = \mathcal{N} \left( \mathbf{x}^N; \mathbf{0}, \mathbf{I} \right)$ back to the data distribution following a Markov chain:

$$p_\theta \left( \mathbf{x}^{i-1} \mid \mathbf{x}^i \right) := \mathcal{N} \left( \mathbf{x}^{i-1}; \boldsymbol{\mu}_\theta \left( \mathbf{x}^i, i \right), \boldsymbol{\Sigma}_\theta \left( \mathbf{x}^i, i \right) \right), \tag{9}$$

where $\theta$ denotes the learnable parameters of Gaussian transitions. Generating data using diffusion models involves sampling starting noises, $\mathbf{x}^N \sim p \left( \mathbf{x}^N \right)$, and implementing reverse process from timestep $i = N$ till $i = 0$.

## 4 Methodology

In this part, we present the details of our DRDT3. Within the unified framework of DRDT3, a diffusion model refines the coarse action predictions from a DT3 module by injecting it as a condition and processing it using a gated MLP noise approximator. A unified objective is proposed to jointly optimize DT3 and the diffusion model.

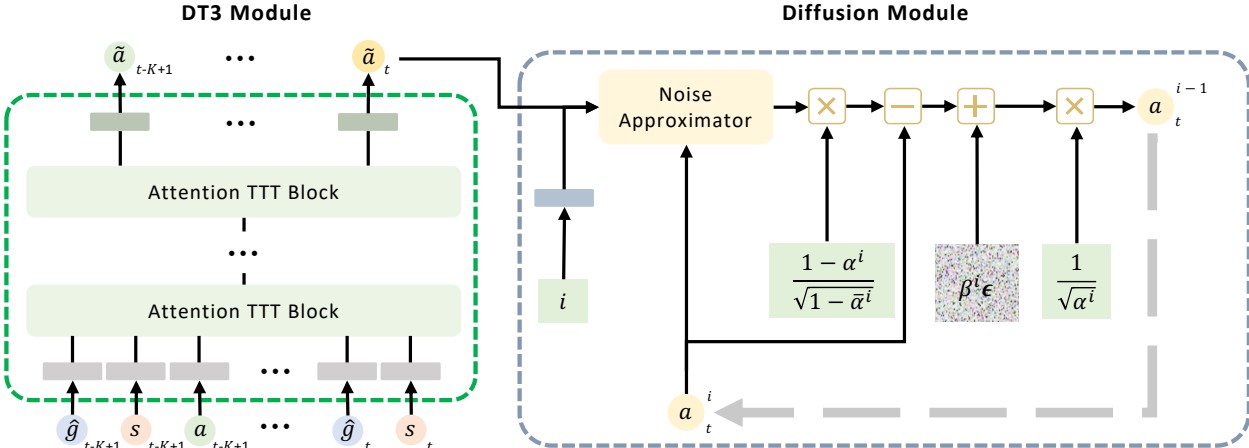

Figure 1: An illustration of the inference procedure of DRDT3. The left and right parts show the structures of DT3 and diffusion modules, respectively. During inference, the coarse action representation predicted by the DT3 module serves as the condition and is refined iteratively within the diffusion module.

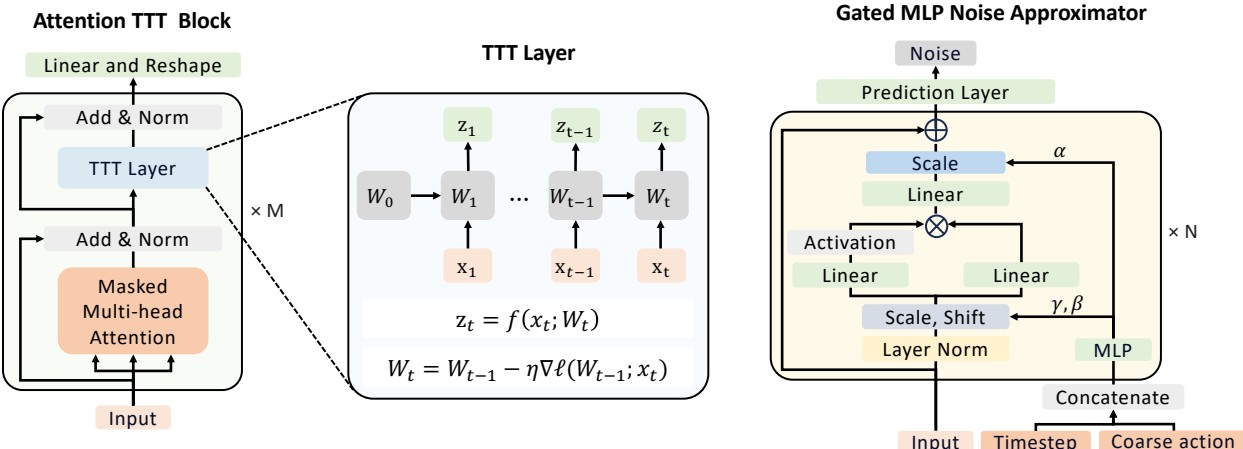

Figure 2: Structure illustration of Attention TTT block, TTT layer, and gated MLP noise approximator.

## 4.1 Decision TTT

The architecture of the DT3 module is depicted in the left part of Figure 1. We modify the standard DT model by replacing Transformer blocks with our proposed Attention TTT blocks for trajectory generation. The structure of the Attention TTT block is detailed in Subsection 4.1.1. In DT3, each input token from the context is projected into the embedding space through a linear layer, and timestep embeddings are subsequently added. Input contexts with a length smaller than $k$ are zero-padded for better generation. With the structure of DT3, actions can be generated autoregressively with known context.

### 4.1.1 Attention TTT Block

Given the competitive performance and linear complexity of TTT layers (Sun et al., 2024) on sequence modelling, we leverage it for trajectory generation in the form of DT. Instead of using the GPT-2 model to process a sequence, we propose an Attention TTT block that exploits both self-attention and the TTT layer. As shown in the left part of Figure 2, we apply masked multi-head attention to capture correlations between input tokens, followed by an Add & Norm layer. We further employ the TTT layer to extract

additional information beyond what is captured by the attention mechanism. The structure of the TTT layer is illustrated in the middle part of Figure 2. In our TTT layer, we use a linear function to represent the weights $W$, resulting in $f(x) = Wx$, where $W$ is a learnable square matrix. The TTT layer is also followed by an Add & Norm layer.

### 4.1.2 Coarse Action Prediction

Besides directly generating the final actions autoregressively using DT3, in our DRDT3, we also leverage the DT3 as one of the modules to predict a coarse representation of the action, which serves as an action condition for the subsequent diffusion model:

$$\widetilde{\boldsymbol{a}}_{-K,t} := \widetilde{\pi}_\theta(\hat{\mathbf{g}}_{-K,t}, \mathbf{s}_{-K,t}), \tag{10}$$

where $\theta$ denotes learnable parameters of DRDT3, $\widetilde{\boldsymbol{a}}_{-K,t}$ is the predicted sequence of actions, $\mathbf{s}_{-K,t}$ and $\hat{\mathbf{g}}_{-K,t}$ are the latest $K$-step RTGs and states, respectively. We use $\widetilde{\boldsymbol{a}}$ to denote the predicted coarse action representation for the current timestep, which is also the final term in $\widetilde{\boldsymbol{a}}_{-K,t}$.

During implementation, we initialize the RTG as the return of the entire trajectory, and the next RTG is calculated using $\hat{g}_t = \hat{g}_{t-1} - r_{t-1}$. When evaluating our DRDT3 online, we introduce an RTG scale factor to adjust the initial RTG. This is achieved by multiplying the maximum positive return from the offline trajectories by $\eta$ while dividing the negative return by it. This defines the desired higher return for the upcoming evaluation.

### 4.2 Conditional Diffusion Policy

We adopt DDPM (Ho et al., 2020), a well-known generative model, as an enhancement module to refine the coarse actions predicted from DT3. To achieve this, we extend DDPM to a conditional diffusion model and employ coarse action $\widetilde{\boldsymbol{a}}$ predicted from DT3 as a prior, as depicted in Figure 1. Consequently, the final action is sampled through a reversed process:

$$\begin{aligned}\pi_\theta(\boldsymbol{a}|\widetilde{\boldsymbol{a}}) &:= p_\theta\left(\boldsymbol{a}^{0:N}|\widetilde{\boldsymbol{a}}\right) \\ &:= p\left(\boldsymbol{a}^N\right) \prod_{i=1}^N p_\theta\left(\boldsymbol{a}^{i-1} \mid \boldsymbol{a}^i, \widetilde{\boldsymbol{a}}\right),\end{aligned} \tag{11}$$

where the final denoised sample $\boldsymbol{a}^0$ denotes the sampled action to be implemented. It is worth noting that we use subscripts $t \in \{1, ..., T\}$ to denote timesteps in trajectories while using superscripts $i \in \{1, ..., N\}$ to represent diffusion timesteps. We formulate $p_\theta\left(\boldsymbol{a}^{i-1} \mid \boldsymbol{a}^i, \widetilde{\boldsymbol{a}}\right)$ using a Gaussian transition

$$\begin{aligned}&p_\theta\left(\boldsymbol{a}^{i-1} \mid \boldsymbol{a}^i, \widetilde{\boldsymbol{a}}\right) \\ &:= \mathcal{N}\left(\boldsymbol{a}^{i-1}; \boldsymbol{\mu}_\theta\left(\boldsymbol{a}^i, \widetilde{\boldsymbol{a}}, i\right), \boldsymbol{\Sigma}_\theta\left(\boldsymbol{a}^i, \widetilde{\boldsymbol{a}}, i\right)\right),\end{aligned} \tag{12}$$

where the covariance matrix is set as $\boldsymbol{\Sigma}_\theta\left(\boldsymbol{a}^i, \widetilde{\boldsymbol{a}}, i\right) = \beta^i \mathbf{I}$. Following DDPM (Ho et al., 2020), we can infer the mean as

$$\boldsymbol{\mu}_\theta\left(\boldsymbol{a}^i, \widetilde{\boldsymbol{a}}, i\right) := \frac{1}{\sqrt{\alpha^i}}\left(\boldsymbol{a}^i - \frac{1-\alpha^i}{\sqrt{1-\bar{\alpha}^i}}\boldsymbol{\epsilon}_\theta\left(\boldsymbol{a}^i, \widetilde{\boldsymbol{a}}, i\right)\right), \tag{13}$$

where the noise prediction model $\boldsymbol{\epsilon}_\theta$ is represented using neural networks. Therefore, during inference, as shown in Algorithm 2, we first sample $\boldsymbol{a}^N \sim \mathcal{N}(\mathbf{0}, \mathbf{I})$. The coarse action can be refined iteratively following the formulation below till we get the final action $\boldsymbol{a}^0$

$$\boldsymbol{a}^{i-1} = \frac{1}{\sqrt{\alpha^i}}\left(\boldsymbol{a}^i - \frac{1-\alpha^i}{\sqrt{1-\bar{\alpha}^i}}\boldsymbol{\epsilon}_\theta\left(\boldsymbol{a}^i, \widetilde{\boldsymbol{a}}, i\right)\right) + \beta^i \boldsymbol{\epsilon}, \tag{14}$$

where $\boldsymbol{\epsilon} \sim \mathcal{N}(\mathbf{0}, \mathbf{I})$ when $i > 1$ while $\boldsymbol{\epsilon} = 0$ when $i = 1$ for better sample quality (Ho et al., 2020). We choose to predict the noise $\boldsymbol{\epsilon}$ using neural networks. Therefore, according to DDPM (Ho et al., 2020), the noise approximator $\boldsymbol{\epsilon}_\theta$ can be optimized with a simplified loss function

$$\mathcal{L}_{diff} := \mathbb{E}_{\boldsymbol{\epsilon}, \boldsymbol{a}, \widetilde{\boldsymbol{a}}, i}\left[\left\|\boldsymbol{\epsilon} - \boldsymbol{\epsilon}_\theta\left(\sqrt{\bar{\alpha}^i}\boldsymbol{a} + \sqrt{1-\bar{\alpha}^i}\boldsymbol{\epsilon}, \widetilde{\boldsymbol{a}}, i\right)\right\|^2\right], \tag{15}$$

---

**Algorithm 1** Training of DRDT3

---

1: Initialize the parameters of policy $\pi_\theta$
2: **repeat**
3:     Sample $\{(\hat{\mathbf{g}}_{-K}, \mathbf{s}_{-K}, \boldsymbol{a}_{-K})\}$ from $D_{offline}$
4:     Predict coarse action representations $\widetilde{\boldsymbol{a}}_{-K} = \widetilde{\pi}_\theta(\hat{\mathbf{g}}_{-K}, \mathbf{s}_{-K})$ and extract $\widetilde{\boldsymbol{a}}$
5:     Compute $\mathcal{L}_{dt3}$ by Equation 16                                       ▷ DT loss
6:     Sample $i \sim \mathcal{U}(1, N)$
7:     Sample $\boldsymbol{\epsilon} \sim \mathcal{N}(\mathbf{0}, \mathbf{I})$
8:     Compute $\mathcal{L}_{diff}$ by Equation 15                                       ▷ Diffusion loss
9:     Update policy by minimizing $\mathcal{L}_{DRDT3} = \mathcal{L}_{diff}(\theta) + \zeta\mathcal{L}_{dt3}(\theta)$        ▷ Unified objective
10: **until** converged

---

**Algorithm 2** Inference of DRDT3

---

1: Sample $\boldsymbol{a}^N \sim \mathcal{N}(\mathbf{0}, \mathbf{I})$
2: **for** $i = N, ...1$ **do**
3:     Predict coarse action representation $\widetilde{\boldsymbol{a}}$ with the DT3 module        ▷ Coarse action prediction
4:     Sample $\boldsymbol{\epsilon} \sim \mathcal{N}(\mathbf{0}, \mathbf{I})$ if $i > 1$ else $\boldsymbol{\epsilon} = 0$
5:     $\boldsymbol{a}^{i-1} = \frac{1}{\sqrt{\alpha^i}}\left(\boldsymbol{a}^i - \frac{1-\alpha^i}{\sqrt{1-\bar{\alpha}^i}}\boldsymbol{\epsilon}_\theta\left(\boldsymbol{a}^i, \widetilde{\boldsymbol{a}}, i\right)\right) + \beta^i\boldsymbol{\epsilon}$        ▷ Action refinement
6: **end for**
7: Return $\boldsymbol{a}^0$ as the selected action                                       ▷ Final action

---

where $i \sim \mathcal{U}(1, N)$ and $\boldsymbol{a} \sim \mathcal{D}_{offline}$.

### 4.3 Gated MLP Noise Approximator

To fully integrate coarse action predictions into the diffusion model and improve noise prediction accuracy, we introduce a gated MLP noise approximator, as depicted in the right part of Figure 2. The gated MLP noise approximator utilizes adaptive layer normalization (adaLN) (Peebles & Xie, 2023) to capture condition-specific information. It specifically learns the scale and shift parameters, $\gamma$ and $\beta$, for layer normalization, as well as a scaling parameter $\alpha$ based on the concatenated condition of the diffusion timestep and coarse action prediction. Additionally, we employ a gated MLP block (Dauphin et al., 2017; De et al., 2024) to extract multi-granularity information from the input, enhancing the model's noise reconstruction capabilities. This architecture splits the output from the Scale & Shift layer into two separate pathways, each expanding the feature dimensionality by a factor of $M$ using a linear layer. The GeLU (Hendrycks & Gimpel, 2016) non-linearity is applied in one branch, after which the two streams are combined through element-wise multiplication. A final linear layer reduces back the dimensionality, and a residual connection is incorporated to facilitate better gradient flow.

### 4.4 Unified optimization objective

The diffusion loss $\mathcal{L}_{diff}(\theta)$ facilitates refinement for the coarse action representation $\widetilde{\boldsymbol{a}}$ and behavior cloning towards the dataset, while it fails to constrain the DT3 module, potentially leading to deviations in coarse action predictions from the data distribution. To alleviate such issues and jointly optimize DT3 and the diffusion model, we propose a unified optimization objective. This involves introducing a DT3 loss as additional guidance during the training of the diffusion model

$$\mathcal{L}_{dt3} := \frac{1}{K\boldsymbol{a}_{\max}}\mathbb{E}_{(\boldsymbol{a}, \mathbf{s}, \hat{\mathbf{g}})\sim\mathcal{D}_{offline}}\left[\|\boldsymbol{a}_{-K} - \widetilde{\pi}_\theta(\hat{\mathbf{g}}_{-K}, \mathbf{s}_{-K})\|_1\right], \tag{16}$$

where $\boldsymbol{a}_{-K}$ is the ground truth of the latest $K$-step actions, and $\boldsymbol{a}_{\max}$, the maximum action in the action space, serves as a scaling factor to render the $L_1$ loss dimensionless. We choose $L_1$ loss over $L_2$ loss since the former is more robust to outliers and provides stable gradients within the unified objective in Equation

17, without weakening the influence of the diffusion loss. The directional information provided by $L_1$ loss further enhances the effectiveness of coarse-to-fine refinement. It is worth noting that the diffusion model only conditions on the last term $\widetilde{a}$ within the sequence $\widetilde{a}_{-K,t}$, whereas the DT3 loss is computed for the entire sequence.

The final unified loss for DRDT3 is a linear combination of the DT3 loss (action representation term) and diffusion loss (action refinement term)

$$\mathcal{L}_{DRDT3} := \mathcal{L}_{diff}(\theta) + \zeta \mathcal{L}_{dt3}(\theta). \tag{17}$$

where $\zeta$ is a loss coefficient for balancing the two loss terms. Note that the DT3 and diffusion modules are jointly trained in a single stage by minimizing $\mathcal{L}_{\mathrm{DRDT3}}$.

## 4.5  Implementation Details

Algorithm 1 and 2 illustrate the training and inference procedures of DRDT3. When implementing our proposed DRDT3, we train it for 50 epochs with 2000 gradient updates per epoch. The learning rate and batch size are designated as 0.0003 and 2048, respectively. To proceed with historical subtrajectories with DT3 module, we set the context length as 6. The Attention TTT block used in the DT3 module consists of 1-layer self-attention and 1-layer TTT with embedding dimensions of 128. We use a linear layer to output a deterministic coarse representation of the action. To enhance computational efficiency without sacrificing performance, the number of diffusion timesteps is set to 5. We set the variance schedule according to Variance Preserving (VP) SDE (Xiao et al., 2021):

$$\beta^i = 1 - \alpha^i = 1 - e^{-\beta_{\min}\left(\frac{1}{N}\right) - 0.5(\beta_{\max} - \beta_{\min})\frac{2i-1}{N^2}}. \tag{18}$$

Most of the hyperparameters are selected through the Optuna hyperparameter optimization framework (Akiba et al., 2019).

## 5  Experiments

We conduct experiments to evaluate our proposed DRDT3 on the commonly used D4RL benchmark (Fu et al., 2020) using an AMD Ryzen 7 7700X 8-Core Processor with a single NVIDIA GeForce RTX 4080 GPU.

### 5.1  Experimental Settings

#### 5.1.1  Tasks and Datasets

We adopt four Mujoco locomotion tasks from Gym, which are HalfCheetah, Hopper, Walker2D, and Ant, and a goal-reaching task, AntMaze. We further evaluate our methods on more challenging tasks, including the Pen and Door manipulation tasks from the Adroit benchmark, which involve a 24-DoF robot hand, as well as Humanoid locomotion tasks from the Gym environment featuring 348-dimensional observations and 17-dimensional actions. For the Gym tasks, dense rewards are provided, and we use three different dataset settings from D4RL (Fu et al., 2020): Medium (m), Medium-Replay (mr), and Medium-Expert (me). Specifically, Medium datasets contain one million samples collected using a behavior policy that achieves 1/3 scores of an expert policy; Medium-Reply datasets contain the replay buffer during training a policy till reaching the Medium score; Medium-Expert datasets consist of two million samples evenly generated from a medium agent and an expert agent. The AntMaze task aims to move an ant robot to reach a target location with sparse rewards (1 for reaching the goal, otherwise 0). Following experimental settings in a DT variant (Zheng et al., 2022), we adopt the umaze and umaze-diverse datasets for evaluation. For the Pen and Door tasks, we use the human and cloned datasets from D4RL. For the Humanoid tasks, we adopt three offline datasets provided by Bhargava et al. (2023): Medium, Medium-Expert, and Expert.

#### 5.1.2  Baselines

To verify the effectiveness of our proposed DRDT3, we compare it with several baseline methods.

Table 1: Normalized scores of different offline RL methods on Gym and AntMaze tasks from D4RL benchmark. Results of DRDT3 represent the mean and variance across three seeds. The best scores are highlighted in bold.

| Dataset | Environment | BC | TD3+BC | CQL | IQL | DT | ConDT | QDT | EDT | DRDT3 (ours) |
|---|---|---|---|---|---|---|---|---|---|---|
| Medium | HalfCheetah | 42.6 | **48.3** | 44.0 | 47.4 | 42.6 | 43.0 | 42.3 | 42.5 | 44.1±1.3 |
| | Hopper | 52.9 | 59.3 | 58.5 | 66.3 | 67.6 | 74.5 | 66.5 | 63.5 | **92.7±8.8** |
| | Walker2d | 75.3 | 83.7 | 72.5 | 78.3 | 74.0 | 72.8 | 67.1 | 72.8 | **84.2±1.8** |
| Medium-Replay | HalfCheetah | 36.6 | 44.6 | **45.5** | 44.2 | 36.6 | 40.9 | 35.6 | 37.8 | 42.3±0.5 |
| | Hopper | 18.1 | 60.9 | 95.0 | 94.7 | 82.7 | **95.1** | 52.1 | 89.0 | 92.7±2.1 |
| | Walker2d | 26.0 | 81.8 | 77.2 | 73.9 | 66.6 | 72.5 | 58.2 | 74.8 | **85.0±1.8** |
| Medium-Expert | HalfCheetah | 55.2 | 90.7 | 91.6 | 86.7 | 86.8 | 92.6 | 79.0 | 89.1 | **94.2±2.1** |
| | Hopper | 52.5 | 98.0 | 105.4 | 91.5 | 107.6 | 110.3 | 94.2 | 108.7 | **112.5±1.5** |
| | Walker2d | 107.5 | **110.1** | 108.8 | 109.6 | 108.1 | 109.1 | 101.7 | 106.2 | 104.5±6.4 |
| **Average** | | 51.9 | 75.3 | 77.6 | 77.0 | 74.7 | 79.0 | 66.3 | 76.0 | **83.6** |
| Medium | Ant | - | - | - | 99.9 | 93.6 | - | - | 97.9 | **103.3±5.8** |
| Medium-Replay | Ant | - | - | - | 91.2 | 89.1 | - | - | 92.0 | **96.7±5.1** |
| **Average** | | - | - | - | 95.6 | 91.4 | - | - | 95.0 | **100.0** |
| Umaze | Antmaze | 54.6 | 78.6 | 74.0 | **87.5** | 59.2 | - | 67.2 | - | 76.6±1.9 |
| Umaze-Diverse | Antmaze | 45.6 | 71.4 | **84.0** | 62.2 | 53.0 | - | 62.1 | - | 82.5±6.3 |
| **Average** | | 50.1 | 75.0 | 79.0 | 74.9 | 56.1 | - | 64.7 | - | **79.6** |

### *Classic offline RL algorithms:*

• **Behavior Cloning (BC)**: A supervised learning approach that trains a policy to mimic the behavior of expert demonstrations by learning from state-action pairs.

• **TD3+BC** (Fujimoto & Gu, 2021): It adapts the TD3 method (Fujimoto et al., 2018) by incorporating a behavior cloning regularization term into the policy objective. This modification aims to prevent the policy from generating actions that deviate significantly from the offline dataset distribution.

• **Conservative Q-Learning (CQL)** (Kumar et al., 2020): It penalizes the Q-values associated with out-of-distribution actions, thereby encouraging more conservative Q-function estimates.

• **Implicit Q-Learning (IQL)** (Kostrikov et al., 2021): IQL approximates the upper expectile of Q-value distributions, allowing it to learn policies focusing on actions within the dataset distribution.

### *DT-based algorithms:*

• **DT** (Chen et al., 2021): DT formulates the policy learning as a sequence modelling problem by autoregressively generating trajectories using GPT models.

• **Contrastive DT (ConDT)** (Konan et al., 2023): It harnesses the power of contrastive representation learning and return-dependent transformations to cluster the input embeddings based on their associated returns.

• **Q-learning DT (QDT)** (Yamagata et al., 2023): It improves the stitching ability of DT by introducing dynamic programming-derived value function.

• **Elastic DT (EDT)** (Wu et al., 2023): It dynamically adjusts the length of the input context according to the quality of the previous trajectory.

### *Diffusion model-based RL methods:*

• **Synther** (Lu et al., 2023): A diffusion model-based data augmentation method that directly samples synthetic offline data. We use TD3+BC as the evaluation algorithm on the augmented datasets.

• **Diffuser** (Janner et al., 2022): A diffusion-based algorithm that adopts diffusion models as a planner for generating trajectories in the form of model-based offline RL.

• **Decision Diffuser (DD) (Ajay et al., 2023)**: A sequence modeling method that samples future states with a diffusion model and infers actions via inverse dynamics.

Table 2: Performance comparison of DRDT3 and diffusion model-based methods on Gym tasks. While DRDT3 does not surpass the baselines on individual tasks, it achieves the highest average normalized score across all tasks.

| Dataset | Environment | SynthER | Diffuser | DD | AdaptDiffuser | EDP | DRDT3 (ours) |
|---|---|---|---|---|---|---|---|
| Medium | HalfCheetah | 48.3±0.0 | 43.8±0.1 | 45.3±0.3 | 44.3±0.2 | **50.8±0.0** | 44.1±1.3 |
| | Hopper | 51.9±0.1 | 89.5±0.7 | **98.2±0.1** | 95.5±1.1 | 72.6±0.2 | 92.7±8.8 |
| | Walker2d | **86.6±0.0** | 79.4±1.0 | 79.6±0.9 | 83.8±1.1 | 86.5±0.2 | 84.2±1.8 |
| Medium-Replay | HalfCheetah | 43.4±0.0 | 36.0±0.7 | 42.9±0.1 | 36.7±0.8 | **44.9±0.4** | 42.3±0.5 |
| | Hopper | 24.7±0.1 | 91.8±0.5 | **99.2±0.2** | 91.2±0.1 | 83.0±1.7 | 92.7±2.1 |
| | Walker2d | **88.6±0.4** | 58.3±1.8 | 75.6±0.6 | 82.9±1.5 | 87.0±2.6 | 85.0±1.8 |
| Medium-Expert | HalfCheetah | 94.8±0.0 | 90.3±0.1 | 88.9±1.9 | 90.4±0.1 | **95.8±0.1** | 94.2±2.1 |
| | Hopper | 76.6±0.4 | 107.2±0.9 | 110.4±0.6 | 109.3±0.3 | 110.8±0.4 | **112.5±1.5** |
| | Walker2d | 110.0±0.0 | 107.4±0.1 | 108.4±0.1 | 107.7±0.1 | **110.4±0.0** | 104.5±6.4 |
| **Average** | | 69.4 | 78.2 | 83.2 | 82.4 | 82.4 | **83.6** |

Table 3: Performance comparison on challenging robot manipulation tasks, i.e., Pen and Door, and the Humanoid task with high-dimensional observations.

| Tasks | BC | CQL | DT | DRDT3 (ours) |
|---|---|---|---|---|
| pen-human-v1 | 25.8 | 35.2 | 72.4 | 78.3±5.7 |
| pen-cloned-v1 | 38.3 | 27.2 | 40.2 | **57.0±19.8** |
| door-human-v1 | 0.5 | 9.9 | 9.2 | **19.8±6.3** |
| door-cloned-v1 | -0.1 | 0.4 | 3.7 | **16.6±5.2** |
| **Average** | 16.1 | 18.2 | 31.4 | **42.9** |
| humanoid-medium | 13.2 | **49.5** | 23.7 | 45.9±4.4 |
| humanoid-medium-expert | 13.7 | 54.1 | 52.6 | **62.2±4.2** |
| humanoid-expert | 24.2 | **63.7** | 53.4 | 59.1±2.0 |
| **Average** | 17.0 | **55.8** | 43.2 | **55.7** |

- **AdaptDiffuser (Liang et al., 2023)**: It improves the diffusion model-based planner by self-evolving with filtered high-quality synthetic demonstrations.

- **EDP (Kang et al., 2023)**: A diffusion-based policy gradient method that accelerates training by approximating actions from corrupted samples in a single step.

The best scores reported in papers of these baselines are adopted. For ConDT, we calculate the normalized score according to the returns reported in (Konan et al., 2023). For diffusion-based methods, we adopt reproduced results in CleanDiffuser (Dong et al., 2024).

## 5.2 Performance Comparison

Table 1 presents the normalized scores (Fu et al., 2020) of our DRDT3, classic offline RL and DT-based baselines on several tasks from the D4RL benchmark. As we can see, DRDT3 significantly improves the average scores on both Gym and AntMaze tasks. Specifically, DRDT3 outperforms other offline RL methods, including the state-of-the-art DT variants, on 7 out of 11 Gym tasks. For the other 4 Gym tasks, DRDT3 is still comparable over the best baselines. Particularly, the Medium-Expert dataset is not reported for the Ant task, given the lack of comparing baselines evaluated on it. For AntMaze tasks, DRDT3 shows comparable performance over the best baselines for each single task while achieving the best overall results. When compared with the vanilla DT, our DRDT3 presents significant improvements on 12 out of 13 tasks, especially on tasks using Medium and Medium-Replay datasets with sub-optimal trajectories, which demonstrates the enhanced stitching ability of our DRDT3 over DT.

We further evaluate DRDT3 by comparing it with recent diffusion model-based methods. As shown in Table 2, DRDT3 achieves superior performance on only one individual task, yet maintains consistently strong results across all tasks, with only a small gap to the best-performing method. Overall, it attains the highest average normalized score among all diffusion-based methods, underscoring its robustness. This outcome may

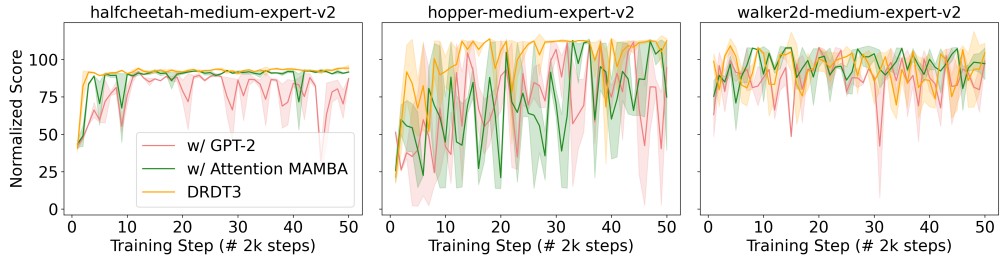

Figure 3: Ablation study on sequence model.

Table 4: Performance comparison between DT, DT3 and DRDT3 on Gym locomotion tasks. The underlined values indicate performance better than DT, and bold values are the best results. 'hc': halfcheetah; 'hp': hopper; 'wk': walker2d.

| | hc-m | hp-m | wk-m | hc-mr | hp-mr | wk-mr | hc-me | hp-me | wk-me | Average |
|---|---|---|---|---|---|---|---|---|---|---|
| DT | 42.6 | 67.6 | 74.0 | 36.6 | 82.7 | 66.6 | 86.8 | 107.6 | **108.1** | 74.7 |
| DT3 (ours) | 42.6±0.7 | 73.0±6.6 | 79.3±1.1 | 39.9±0.4 | 75.9±4.4 | 63.9±2.8 | 93.4±2.7 | 109.9±1.8 | 104.3±8.0 | 75.8 |
| DRDT3 (ours) | **44.1±1.3** | **92.7±8.8** | **84.2±1.8** | **42.3±0.5** | **92.7±2.1** | **85.0±1.8** | **94.2±2.1** | **112.5±1.5** | 104.5±6.4 | **83.6** |

stem from the fact that DRDT3 is inherently constrained by the performance of its underlying DT module, as it directly refines DT-generated actions without incorporating dynamic programming. Consequently, DRDT3's performance is steady rather than outstanding on individual tasks.

Table 3 presents the evaluation performance of DRDT3 on more challenging tasks, including the Pen and Door tasks from the Adroit benchmark and the Humanoid task from Gym. We report results for BC, CQL, and DT as baselines, since other methods are rarely evaluated on these tasks. As shown, DRDT3 consistently outperform the baseline methods on most Pen and Door datasets, with DRDT3 achieving an average improvement of 36.6% over DT. For the Humanoid task, DRDT3 attains performance comparable to the best baseline, CQL, while still delivering a 28.9% improvement over DT. The underperformance observed on certain tasks likely stems from an inherent limitation of sequence modeling methods, which lack dynamic programming capabilities to effectively extrapolate toward higher performance.

Our primary goal, however, is to demonstrate that diffusion models can effectively refine DT-generated actions and enhance DT's trajectory-stitching capability—an improvement our experiments confirm across a wide range of DT-based baselines and evaluation tasks. In particular, DRDT3 achieves average improvements of 11.9%, 9.4%, 41.9%, 36.6%, and 28.9% over DT on Gym locomotion, Ant, AntMaze, Adroit, and Humanoid tasks, respectively.

## 5.3 Ablation on Sequence Model

We compare our DRDT3 framework with two variants: one where the sequence model is replaced by a GPT-2 model (Radford et al., 2019) and another where the TTT layer is replaced by an MAMBA layer (Gu & Dao, 2023). The GPT-2 model relies entirely on the attention mechanism, while MAMBA is a selective state space model based on the RNN structure, where parameters are dynamically dependent on the inputs. For the two variants w/GPT-2 and w/ Attention MAMBA, we use one block per model as our DRDT3. To ensure a fair comparison, we adjust the number of layers in both the GPT-2 and MAMBA variants to match

Table 5: Performance comparison between DT3 and DT on Ant and AntMaze tasks.

| | ant-m | ant-mr | Average | antmaze-umaze | antmaze-umaze-diverse | Average |
|---|---|---|---|---|---|---|
| DT | 93.6 | 89.1 | 91.4 | 59.2 | 53.0 | 56.1 |
| DT3 (ours) | 96.7±7.3 | 89.0±5.2 | 92.9 | 62.4±3.3 | 53.0±4.8 | 57.7 |
| DRDT3 (ours) | **103.3±5.8** | **96.7±5.1** | **100.0** | **76.6±1.9** | **82.5±6.3** | **79.6** |

Table 6: Performance comparison of DT, DT3, and DRDT3 across Adroit and Humanoid tasks.

| Method | pen-h | pen-c | door-h | door-c | Avg. | human-m | human-me | human-e | Avg. |
|--------|-------|-------|--------|--------|------|---------|----------|---------|------|
| DT | 72.4 | 40.2 | 9.2 | 3.7 | 31.4 | 23.7 | 52.6 | 53.4 | 43.2 |
| DT3 (ours) | **79.0±2.2** | 38.9±20.1 | 12.7±6.2 | 3.7±1.8 | 33.6 | 29.2±5.8 | 51.5±2.2 | 54.0±2.3 | 44.9 |
| DRDT3 (ours) | 78.3±5.7 | **57.0±19.8** | **19.8±6.3** | **16.6±5.2** | **42.9** | **45.9±4.4** | **62.2±4.2** | **59.1±2.0** | **55.7** |

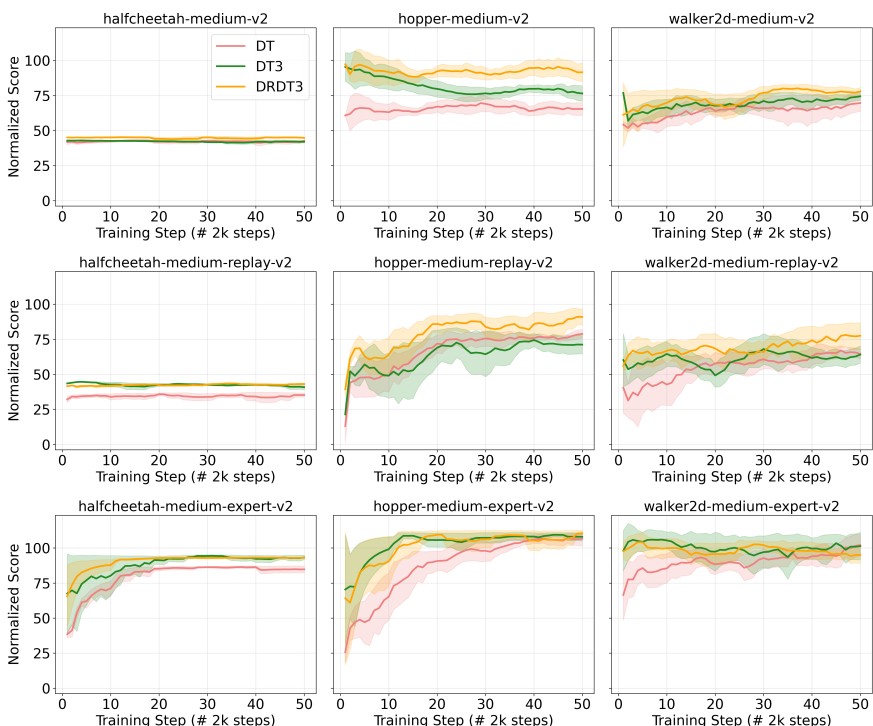

Figure 4: Learning curves of DT, DT3, and DRDT3 on Gym tasks. For clarity, each curve is plotted with a moving average (window size = 10) to highlight the trend.

the model size of our DRDT3. Specifically, the total number of learnable parameters for our DRDT3, the variant w/GPT-2, and the variant w/ Attention MAMBA are 235K, 218K, and 329K, respectively.

As shown in Figure 3, our DRDT3 with Attention TTT blocks outperforms the other variants across three Medium-Expert tasks. The variant with Attention MAMBA shows slight performance degradation and lacks the stability of DRDT3. The variant with GPT-2 performs the worst across all tasks, which might be attributed to the typical reliance on larger model sizes of GPT-2. When constrained to a similar size as the other two variants, GPT-2 experiences performance degradation. These results indicate that our proposed DT3 module with Attention TTT blocks enhances performance without increasing model size.

### 5.4 Ablation on Diffusion

To test whether the diffusion module plays a significant role in action refinement and verify the effectiveness of our DT3 module, we train a DT3 model along with the same settings as DRDT3 by minimizing the $\mathcal{L}_{dt3}$ in Equation 16. Table 4, 5, and 6 show the performance comparison between DT and our DT3 model on Gym, AntMaze, and challenging robotic (Adroit and Humanoid) tasks. Figure 4 illustrates the learning curves of the three methods on Gym locomotion tasks.

As we can see, our proposed DT3 trained independently can outperform DT on 6 out of 9 Gym locomotion tasks, all AntMaze tasks, 3 out of 4 Adroit tasks, and 2 out of 3 Humanoid tasks. This demonstrates the superiority of our proposed DT3 with the Attention TTT blocks over the vanilla DT with only the attention

Table 7: Model size and average training and inference time comparison. # param. denotes the number of learnable parameters. Training time refers to the average time required for one epoch of model training, while inference time indicates the time needed to evaluate the policy for a single episode.

| | Model Size | Hopper-Medium | | HalfCheetah-Medium | | Walker2d-Medium | |
| | # param. (K) | Training (s) | Inference (s) | Training (s) | Inference (s) | Training (s) | Inference (s) |
|---|---|---|---|---|---|---|---|
| DT | 226.8 | 162.2 | **1.7** | 258.7 | **2.6** | 239.1 | **2.6** |
| DT3 | 234.6 | **134.3** | 2.6 | **245.8** | 4.2 | **223.9** | 3.1 |

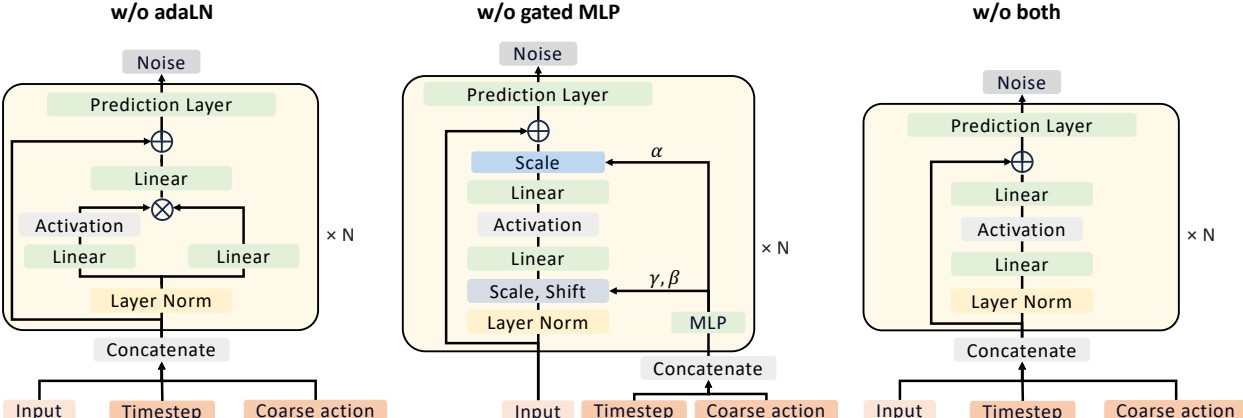

Figure 5: Structure illustration of three noise approximators. **Left**: w/o adaLN; **Middle**: w/o gated MLP; **Right**: w/o both.

mechanism. Furthermore, our proposed DRDT3, which enhances DT3 through a diffusion module, further boosts performance across all tasks with different datasets, achieving average improvements of 10.3%, 7.6%, 38.0%, 27.7%, and 24.1% on Gym locomotion, Ant, AntMaze, Aroit, and Humanoid tasks, respectively. This demonstrates the substantial impact of incorporating diffusion refinement in DRDT3.

However, underperformance is observed in the walker2d-me task. As shown in Figure 4, we can see a slight degradation in the learning curve as training iterations increase, indicative of overfitting to the dataset's mixed-quality dynamics. Specifically, as training progresses, the DT3 module might memorize noisy or erratic details from medium-quality trajectories rather than generalizing to the smooth, energy-efficient patterns of expert demonstrations. This leads to increasingly misaligned coarse actions, which the diffusion module cannot fully correct. These may result in degraded policy performance on this task.

## 5.5 Computational Efficiency

We evaluate the computational efficiency of our proposed DT3, which consists of Attention–TTT blocks, by comparing it against DT, which composed solely of Attention blocks. Table 7 reports the model sizes along with the average training and inference times. As shown, DT3 consistently achieves faster training but fails to surpass DT in inference speed. This is consistent with the findings in the original TTT paper (Sun et al., 2024), which reported that TTT layers offer a clear advantage over Transformers during training and become more efficient at inference when processing long contexts. In our case, the sequence length is only 6, a relatively short context, which prevents DT3 from benefiting from TTT's inference-time speed advantage.

## 5.6 Ablation on Noise Approximator

We also verify the effectiveness of our gated MLP noise approximator by comparing DRDT3 with several variants, as shown in Figure 5: one where the gated MLP structure is replaced by a simple MLP, another where the adaLN is replaced with in-context conditioning (Peebles & Xie, 2023), and a third variant that

Table 8: Ablation studies on the *noise approximator* (left side) and *unified loss function* (right side).

| Dataset | w/o adaLN | w/o gated mlp | w/o both | DRDT3 (ours) | w/o $\mathcal{L}_{dt3}$ | w/ L2 $\mathcal{L}_{dt3}$ |
|---|---|---|---|---|---|---|
| halfcheetah-m | 43.4±1.6 | 44.0±1.5 | 43.8±1.2 | **44.1±1.3** | **44.1±1.4** | 42.5±1.3 |
| hopper-m | 91.3±4.9 | 77.3±5.2 | 75.4±6.0 | **92.7±8.8** | 81.2±8.2 | 92.0±5.2 |
| walker2d-m | 84.0±1.0 | 83.4±1.1 | 82.5±1.8 | **84.2±1.8** | 80.5±2.1 | 79.8±1.2 |
| halfcheetah-mr | 42.7±0.8 | 42.8±0.9 | **44.4±0.8** | 42.3±0.5 | 42.7±0.9 | 38.2±0.9 |
| hopper-mr | **92.7±1.3** | 86.9±1.5 | 85.9±5.1 | **92.7±2.1** | 84.2±5.4 | 91.5±1.7 |
| walker2d-mr | 85.5±2.8 | 82.2±2.7 | 80.7±0.9 | 85.0±1.8 | 85.1±1.3 | **86.0±2.9** |
| halfcheetah-me | 93.8±2.5 | 89.8±3.1 | 90.3±2.5 | **94.2±2.1** | 92.3±2.6 | 93.3±3.0 |
| hopper-me | 111.9±1.7 | 106.6±1.5 | 105.9±1.3 | **112.5±1.5** | 105.1±1.6 | 110.3±1.5 |
| walker2d-me | 102.6±1.3 | 101.1±1.1 | 98.4±5.2 | **104.5±6.4** | 100.7±5.6 | 103.0±1.3 |
| **Average** | 83.1 | 79.3 | 78.6 | **83.6** | 79.5 | 81.8 |

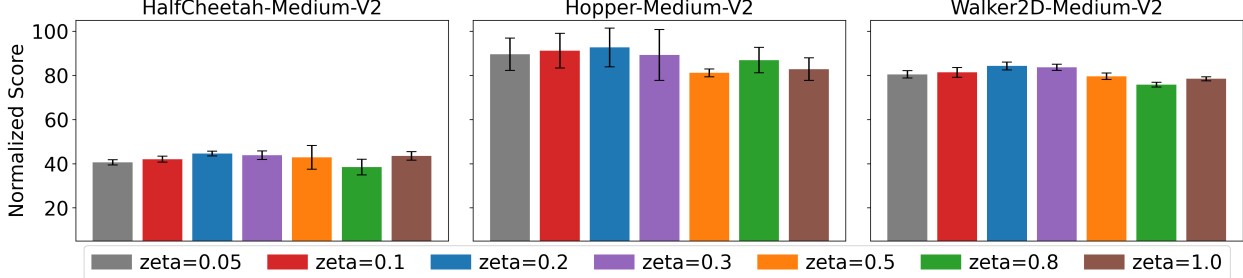

Figure 6: Sensitivity analysis on the loss coefficient $\zeta$. We empirically choose $\zeta = 0.2$ according to the performance comparison.

removes both structures. As shown in Table 8, the removal of adaLN causes a slight performance degradation, while removing the gated MLP results in a significant reduction in performance.

### 5.7 Ablation on Loss Function

To verify the effectiveness of our proposed unified optimization objective, we compare our loss function with the vanilla diffusion loss function (w/o $\mathcal{L}_{dt3}$) and a unified loss function incorporating an L2-based DT3 loss term (w/ L2 $\mathcal{L}_{dt3}$). As shown in Table 8, the variant w/o $\mathcal{L}_{dt3}$ shows worse performance than the variant with an extra L2-based DT3 constraint. Our DRDT3, utilizing an L1-based DT3 loss term instead, can further enhance performance beyond the L2 variant. These results demonstrate that incorporating DT3 loss into the final loss improves performance, with the L1 $\mathcal{L}_{dt3}$ offering additional gains over the L2 variant.

### 5.8 Sensitivity Analysis on Loss Coefficient

We further vary the loss coefficient $\zeta$ across different values to select the best one empirically. Figure 6 illustrates the normalized scores of our DRDT3 with varying $\zeta$ and trained on three Gym tasks with Medium datasets. We empirically set $\zeta = 0.2$ based on the results.

## 6 Conclusion and Discussion

This work introduces the DRDT3 algorithm to enhance the stitching ability of the DT-based method. We propose a novel DT3 module with attention TTT blocks to process historical information and predict a better coarse action representation. The diffusion module then conditions this coarse representation and iteratively refines the action using a novel gated MLP noise approximator, improving performance on sub-optimal datasets. To achieve joint learning of the DT3 and diffusion modules, we introduce a unified optimization objective, consisting of both an action representation term and an action refinement term. Experiments on a variety of tasks with different dataset configurations from the D4RL benchmark demonstrate the effectiveness

and superior performance of DRDT3 compared to conventional offline RL methods and DT-based approaches. Experiment results also demonstrate that DT3 can outperform the vanilla DT.

Nonetheless, we acknowledge that there remains a performance gap between our DRDT3 and certain state-of-the-art dynamic programming–assisted diffusion-based policies. This gap likely arises because DRDT3 is inherently constrained by the performance of the underlying DT3 module, as it directly refines the actions generated by DT3. Therefore, we plan to further investigate strategies to narrow this gap in the future, aiming to bring DT-based methods closer to the performance of their dynamic programming–based counterparts.

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
