# OpenReview forum: "DRDT3: Diffusion-Refined Decision Test-Time Training Model"
_TMLR — Accepted by TMLR_

### Review · Reviewer_2Ywh · 2025-07-03

**Summary Of Contributions:**

This paper introduces DRDT3, a novel offline reinforcement learning framework designed to enhance the Decision Transformer (DT) by improving its ability to learn from suboptimal data, a concept known as trajectory stitching. The core idea is a two-stage process where a base model first predicts a coarse action, which is then iteratively refined by a diffusion model. The base model, named Decision TTT (DT3), replaces the standard transformer blocks in DT with a novel "Attention TTT block." This block combines self-attention with a Test-Time Training (TTT) layer, an RNN-like structure that updates its hidden states at test time, aiming to better capture recent context. This DT3 module generates a coarse action prediction. This coarse action then serves as a condition for a conditional diffusion model. The diffusion model, using a novel gated MLP noise approximator, iteratively denoises a random sample, guided by the coarse action, to produce a final, more refined action. The main contributions are the DT3 model, which improves upon the standard DT, and the full DRDT3 framework, which uses diffusion for action refinement. The authors propose a unified optimization objective to train both the DT3 and diffusion components jointly. Through extensive experiments on D4RL benchmarks, the paper shows that DRDT3 achieves best results among the conventional offline RL methods and other DT-based approaches.

**Audience:**

Yes

**Claims And Evidence:**

Yes

**Requested Changes:**

I highly recommend to update the related work section and experiments substantially, and discuss the relevant to this work and why the performance gap happens there. At least please consider to include the comparison and discussion against the following three (https://arxiv.org/abs/2305.20081; https://arxiv.org/abs/2406.09509; https://arxiv.org/abs/2405.20555). Please also note that these three are not all the relevant works for offline RL with diffusion policy, and survey the prior works extensively.

**Strengths And Weaknesses:**

### Strengths
- [s1] The paper is well-written and is easy to follow.
- [s2] The architectural changes in Decision Transformer, replacing Transformer block with Attention TTT block, sounds interesting. This change makes the model leverage the historical information by RNN-like structure.
- [s3] Two-staged approach, incorporating diffusion models to refine the action, can boost the performance.
- [s4] The paper conducts extensive ablation studies on architecture and loss functions, which would provide insightful information for practitioners.

### Weaknesses
- [w1] I think that the discussion and comparison around the diffusion policy is quite insufficient. While there is a lot of works using diffusion policy in offline RL, no baseline, except for Diffuser, is included in Table 1. For instance, even with my coarse survey, these three papers proposed offline RL methods with diffusion policies, and some achieved better performance than DRDT3 (https://arxiv.org/abs/2305.20081; https://arxiv.org/abs/2406.09509; https://arxiv.org/abs/2405.20555). I understood TMLR is not a venue aiming for SoTA, and not achieving SoTA should not be a reason for rejection, but that does not mean we can ignore the previous works in the literature. I highly recommend to update the related work section and experiments substantially, and discuss the relevant to this work and why the performance gap happens there.
- [w2] it is unclear if DT module needed to be pre-trained, and then optimized through the unified objective in Section 4.4, or if we can train the policy from scratch just by optimizing the unified objective.
- [w3] are there any reasons to use L1 loss in equation 16?

---

> ### Author Response · Authors · 2025-08-14
>
> We sincerely thank you for your insightful comments. We have modified our manuscript accordingly and highlighted all revisions in blue in the updated manuscript.
>
> *1.	[w1] I think that the discussion and comparison around the diffusion policy is quite insufficient. While there is a lot of works using diffusion policy in offline RL, no baseline, except for Diffuser, is included in Table 1. For instance, even with my coarse survey, these three papers proposed offline RL methods with diffusion policies, and some achieved better performance than DRDT3 (https://arxiv.org/abs/2305.20081; https://arxiv.org/abs/2406.09509; https://arxiv.org/abs/2405.20555). I understood TMLR is not a venue aiming for SoTA, and not achieving SoTA should not be a reason for rejection, but that does not mean we can ignore the previous works in the literature. I highly recommend to update the related work section and experiments substantially, and discuss the relevant to this work and why the performance gap happens there.*
>
> **Response**:
>
> We sincerely appreciate the valuable suggestion. In the revised manuscript, we have substantially expanded the related work section, added 13 recent and closely related studies, and provided a more detailed discussion on their connection to our approach. The modified Related Work for Diffusion Models in Reinforcement Learning part is as follows:
>
> Diffusion models (Ho et al., 2020) have been appealing for their strong capacity to generate high-dimensional image or text data. In light of the expressive representation and strong multi-modal distribution modelling ability of the diffusion model, some researchers have introduced the diffusion model to RL paradigms. Diffuser (Janner et al., 2022) first employs diffusion models as a planner for generating trajectories in model-based offline RL, which alleviates the severe compounding errors of conventional planners. AdaptDiffuser (Liang et al., 2023) improves the diffusion model-based planner by self-evolving with filtered high-quality synthetic demonstrations. In addition to employing diffusion models as planners, SynthER (Lu et al., 2023), DiffStich (Li et al., 2024), GODA Huang et al. (2024), and PRIDE Feng et al. further exploit diffusion models as data synthesizers to directly augment either offline or online training data with higher diversity and quality.
>
> Diffusion models have also been adopted to represent policies. Decision Diffuser (Ajay et al., 2023) formulates sequential decision-making problems as conditional generative modelling and introduces constraints and skill conditions. Diffusion-QL (Wang et al., 2022) explores representing policy as a diffusion model and employing Q-value function guidance during training. It overcomes the over-conservatism of policies learned from conventional offline RL. Efficient Diffusion Policy (EDP) (Kang et al., 2023) extends diffusion models to policy gradient methods and accelerates training by approximating actions from corrupted samples in a single step. Diffusion Actor-Critic (DAC) (Fang et al., 2024) adopts an actor-critic training framework, training a critic network alongside a diffusion-based KL-constrained policy, where soft Q-guidance is used to regularize the policy to remain close to the behavior policy. Ma et al. (2025) develop two diffusion-based online RL algorithms, i.e., Diffusion Policy Mirror Descent (DPMD) and Soft Diffusion Actor-Critic (SDAC), built on two novel reweighted score matching methods. MaxEntDP (Dong et al., 2025) and DIME (Celik et al., 2025) incorporate maximum entropy RL to enhance online exploration with diffusion-represented policies. DRCORL (Guo et al., 2025) applies diffusion-based policies to constrained RL and introduces a gradient operation to balance reward and safety constraints. FDPP (Chen et al., 2025) and PRIDE (Feng et al.) further explore the application of diffusion models in preference-based RL. Moreover, CleanDiffuser (Dong et al., 2024) offers a diffusion model library that facilitates the development of diffusion-based decision-making methods.
>
> Our DRDT3 uses a unified framework that aims to enhance a DT-style trajectory modeling method by refining the predictions iteratively within the expressive DDPM. In contrast to the above methods, DRDT3 employs the diffusion model solely as an action refinement module, directly operating on the DT output without multi-stage training. It remains a sequence-modeling approach without dynamic programming, avoiding the compounding errors of dynamics models and value estimation.

---

> ### Author Response · Authors · 2025-08-14
>
> **Continuing from Q1**
>
> We further added experimental results comparison with more diffusion model-based methods in Table 2 in our revised manuscript, including:
>
> • Synther (Lu et al., 2023): A diffusion model-based data augmentation method that directly samples synthetic offline data. We use TD3+BC as the evaluation algorithm on the augmented datasets.
>
> • Diffuser (Janner et al., 2022): A diffusion-based algorithm that adopts diffusion models as a planner for generating trajectories in the form of model-based offline RL.
>
> • Decision Diffuser (DD) (Ajay et al., 2023): A sequence modeling method that samples future states with a diffusion model and infers actions via inverse dynamics.
>
> • AdaptDiffuser (Liang et al., 2023): It improves the diffusion model-based planner by self-evolving with filtered high-quality synthetic demonstrations.
>
> • EDP (Kang et al., 2023): A diffusion-based policy gradient method that accelerates training by approximating actions from corrupted samples in a single step.
>
> We adopt the reproduced results from CleanDiffuser [1].
>
> **Performance comparison of DRDT3 and diffusion model-based methods on Gym tasks.**
> | Dataset         | Environment | SynthER        | Diffuser       | DD             | AdaptDiffuser | EDP            | **DRDT3 (ours)** |
> |-----------------|-------------|----------------|----------------|----------------|---------------|----------------|------------------|
> | **Medium**      | HalfCheetah | 48.3±0.0       | 43.8±0.1       | 45.3±0.3       | 44.3±0.2      | **50.8±0.0**   | 44.1±1.3         |
> |                 | Hopper      | 51.9±0.1       | 89.5±0.7       | **98.2±0.1**   | 95.5±1.1      | 72.6±0.2       | 92.7±8.8         |
> |                 | Walker2d    | **86.6±0.0**   | 79.4±1.0       | 79.6±0.9       | 83.8±1.1      | 86.5±0.2       | 84.2±1.8         |
> | **Medium-Replay** | HalfCheetah | 43.4±0.0       | 36.0±0.7       | 42.9±0.1       | 36.7±0.8      | **44.9±0.4**   | 42.3±0.5         |
> |                 | Hopper      | 24.7±0.1       | 91.8±0.5       | **99.2±0.2**   | 91.2±0.1      | 83.0±1.7       | 92.7±2.1         |
> |                 | Walker2d    | **88.6±0.4**   | 58.3±1.8       | 75.6±0.6       | 82.9±1.5      | 87.0±2.6       | 85.0±1.8         |
> | **Medium-Expert** | HalfCheetah | 94.8±0.0       | 90.3±0.1       | 88.9±1.9       | 90.4±0.1      | **95.8±0.1**   | 94.2±2.1         |
> |                 | Hopper      | 76.6±0.4       | 107.2±0.9      | 110.4±0.6      | 109.3±0.3     | 110.8±0.4      | **112.5±1.5**    |
> |                 | Walker2d    | 110.0±0.0      | 107.4±0.1      | 108.4±0.1      | 107.7±0.1     | **110.4±0.0**  | 104.5±6.4        |
> | **Average**     |             | 69.4           | 78.2           | 83.2           | 82.4          | 82.4           | **83.6**         |
>
> As shown in **Table 2** in our revised manuscript, DRDT3 achieves superior performance on only one individual task, yet maintains consistently strong results across all tasks, with only a small gap to the best-performing method. Overall, it attains the highest average normalized score among all diffusion-based methods, underscoring its robustness. This outcome may stem from the fact that DRDT3 is inherently constrained by the performance of its underlying Decision Transformer (DT) module, as it directly refines DT-generated actions without incorporating dynamic programming. Consequently, DRDT3’s performance is steady rather than outstanding on individual tasks.
>
> Our primary goal, however, is to demonstrate that diffusion models can effectively refine DT-generated actions and enhance DT’s trajectory-stitching capability—an improvement our experiments confirm across a wide range of DT-based baselines.
>
> We also added a limitation discussion in the Conclusion part in Section 6 as follows:
>
> "Nonetheless, we acknowledge that there remains a performance gap between our DRDT3 and certain state-of-the-art dynamic programming–assisted diffusion-based policies. This gap likely arises because DRDT3 is inherently constrained by the performance of the underlying DT3 module, as it directly refines the actions generated by DT3. Therefore, we plan to further investigate strategies to narrow this gap in the future, aiming to bring DT-based methods closer to the performance of their dynamic programming–based counterparts."

---

> ### Author Response · Authors · 2025-08-14
>
> *2.	[w2] it is unclear if DT module needed to be pre-trained, and then optimized through the unified objective in Section 4.4, or if we can train the policy from scratch just by optimizing the unified objective.*
>
> **Response**:
>
> Thanks for pointing out this ambiguity. Our policy is trained from scratch using a unified objective. Unlike QDT, Diffuser, or Decision Diffuser, our approach does not require multi-stage training. Instead, we jointly train the entire model, comprising both the DT and diffusion modules, in a single run under the unified objective. We have clarified this point in the revised manuscript to avoid confusion.
>
> ---
>
> *3.	[w3] are there any reasons to use L1 loss in equation 16?*
>
> **Response**:
>
> Thank you for the question. Using L1 loss in Equation 16 (L_dt3) is motivated by key considerations aligned with offline RL and DT3’s role, supported by empirical validation:
>
> *Robustness to noisy offline data*: Offline datasets often contain suboptimal trajectories or outliers. L1 loss minimizes their influence by penalizing errors linearly, ensuring DT3 focuses on the central tendency of meaningful patterns rather than overfitting to noise and avoiding drift from high-quality data distributions.
>
> *Training stability*: L1 loss provides consistent gradient magnitudes (±1) regardless of error size, avoiding the large, destabilizing updates that L2 loss (with error-scaled gradients) can introduce. This stability is vital for learning coherent long-horizon trajectories amid diverse data.
>
> *Alignment with coarse action goals*: DT3 generates coarse actions as diffusion priors. L1 loss prioritizes "directionally correct" actions (small absolute errors) over minimizing squared deviations, ensuring priors are more meaningful for refinement.
>
> Empirically, our ablation studies (Table 5) confirm that L1 loss in Equation 16 outperforms L2 loss, yielding average performance gains of 2.2% across Gym tasks.
>
> We have added these explanations in our revised version to make it clearer.
>
> ---
>
> We greatly appreciate your insightful comments and suggestions. We hope our responses and revisions have resolved your concerns and improved the clarity of the manuscript.

---

> ### Comment · Reviewer_2Ywh · 2025-08-30
>
> Thank you for updating the manuscripts. Overall, the paper looks good to me and my concern is resolved.

---

### Review · Reviewer_7Lmj · 2025-07-05

**Summary Of Contributions:**

This manuscript considers offline RL problems and introduces two components to enhance algorithm performance: a novel DT-style trajectory modelling method and a conditional diffusion model. This approach enables the predicted actions from DT3 to be iteratively refined with Gaussian noise through the denoising chain within the diffusion model, facilitating trajectory stitching. Experiments on extensive tasks demonstrate the effectiveness of the proposed methodology.

**Audience:**

Yes

**Broader Impact Concerns:**

NO.

**Claims And Evidence:**

Yes

**Requested Changes:**

Please refer to the above weaknesses.

**Strengths And Weaknesses:**

**Strengths**:
- This article is well written.
- The chosen baselines are both comprehensive and appropriate for the evaluation of the proposed method.
- The ablation study validates the effectiveness of the proposed method.

**Weaknesses**:
- Generalization to high-dimensional observations remains unexplored. The tasks in the experiments are relatively simple with low-dimensional state and action spaces. The authors are recommended to add experiments on more complex Mujoco tasks, such as Humanoid and Humanoid Standup, which have a a state-space dimension of 348 and an action-space dimension of 17.
- The authors are recommended to provide training curves of the baselines to compare the method performance during the learning phase.
- As an offline RL approach, the stability of the proposed training strategy is suggested to be discussed theoretically.
- The normalized scores of the methods show little improvement on the walker2d-medium-expert-v2 task, as depicted in Figure 3. The paper could benefit from further discussion of this phenomenon.
- Several references require page numbers to meet standard citation practices:
    -   Takuya Akiba, Shotaro Sano, Toshihiko Yanase, Takeru Ohta, and Masanori Koyama. Optuna: A nextgeneration hyperparameter optimization framework. In Proceedings of the 25th ACM SIGKDD International Conference on Knowledge Discovery and Data Mining, 2019.
    -  Hanqun Cao, Cheng Tan, Zhangyang Gao, Yilun Xu, Guangyong Chen, Pheng-Ann Heng, and Stan Z Li. A survey on generative diffusion models. IEEE Transactions on Knowledge and Data Engineering, 2024.
   -   Siyuan Guo, Lixin Zou, Hechang Chen, Bohao Qu, Haotian Chi, S Yu Philip, and Yi Chang. Sample efficient offline-to-online reinforcement learning. IEEE Transactions on Knowledge and Data Engineering, 2023.
   -  Jia Liu, Yunduan Cui, Jianghua Duan, Zhengmin Jiang, Zhongming Pan, Kun Xu, and Huiyun Li. Reinforcement learning-based high-speed path following control for autonomous vehicles. IEEE Transactions on Vehicular Technology, 2024a.
   -  Cong Lu, Philip Ball, Yee Whye Teh, and Jack Parker-Holder. Synthetic experience replay. Advances in Neural Information Processing Systems, 36, 2024.
   -  Hamid Taghavifar, Chuan Hu, Chongfeng Wei, Ardashir Mohammadzadeh, and Chunwei Zhang. Behaviorally-aware multi-agent rl with dynamic optimization for autonomous driving. IEEE Transactions on Automation Science and Engineering, 2025.
   -  Ashish Vaswani, Noam Shazeer, Niki Parmar, Jakob Uszkoreit, Llion Jones, Aidan N Gomez, Łukasz Kaiser, and Illia Polosukhin. Attention is all you need. Advances in neural information processing systems, 30, 2017.
   -  MaonanWang, XiXiong, YuhengKan, Chengcheng Xu, andMan-OnPun. Unitsa: Auniversal reinforcement learning framework for v2x traffic signal control. IEEE Transactions on Vehicular Technology, 2024a.
   -  Yueh-Hua Wu, Xiaolong Wang, and Masashi Hamaya. Elastic decision transformer. Advances in Neural Information Processing Systems, 36, 2024.
   -  Ziyang Zhai, Ruru Hao, Boyang Cui, and Siyi Wang. Hgat and multi-agent rl-based method for multiintersection traffic signal control. IEEE Transactions on Intelligent Transportation Systems, 2025.

---

> ### Author Response · Authors · 2025-08-14
>
> We sincerely thank you for your insightful comments. We have modified our manuscript accordingly and highlighted all revisions in blue in the updated manuscript.
>
> *1.	Generalization to high-dimensional observations remains unexplored. The tasks in the experiments are relatively simple with low-dimensional state and action spaces. The authors are recommended to add experiments on more complex Mujoco tasks, such as Humanoid and Humanoid Standup, which have a state-space dimension of 348 and an action-space dimension of 17.*
>
> **Response**:
>
> Thanks for the valuable suggestions. We checked for the available public offline datasets for these two tasks and could only find available datasets for Humanoid [1]. Therefore, we adopt this task for evaluation. To make our experiments more comprehensive, we also add experiments on complex robot manipulation tasks: Pen and Door from the Adroit benchmark.
>
> | **Tasks**                    | BC    | CQL   | DT    | DT3 (ours)       | DRDT3 (ours)     |
> |------------------------------|-------|-------|-------|------------------|------------------|
> | pen-human-v1                 | 25.8  | 35.2  | 72.4  | **79.0±2.2**     | 78.3±5.7         |
> | pen-cloned-v1                 | 38.3  | 27.2  | 40.2  | 38.9±20.1        | **57.0±19.8**    |
> | door-human-v1                  | 0.5   | 9.9   | 9.2   | 12.7±6.2         | **19.8±6.3**     |
> | door-cloned-v1                 | -0.1  | 0.4   | 3.7   | 3.7±1.8          | **16.6±5.2**     |
> | **Average**                   | 16.1  | 18.2  | 31.4  | 33.6             | **46.4**         |
> | humanoid-medium               | 13.2  | **49.5** | 23.7  | 29.2±5.8         | 45.9±4.4         |
> | humanoid-medium-expert        | 13.7  | 54.1  | 52.6  | 51.5±2.2         | **62.2±4.2**     |
> | humanoid-expert               | 24.2  | **63.7** | 53.4  | 54.0±2.3         | 59.1±2.0         |
> | **Average**                   | 17.0  | **55.8** | 43.2  | 44.9             | **55.7**         |
>
> Table 3 in our revised manuscript presents the evaluation performance of DRDT3 on more challenging tasks, including the Pen and Door tasks from the Adroit benchmark and the Humanoid task from Gym. We report results for BC, CQL, and DT as baselines, since other methods are rarely evaluated on these tasks.
>
> As shown, both DT3 and DRDT3 consistently outperform the baseline methods on most Pen and Door datasets, with DRDT3 achieving an average improvement of 47.8% over DT. For the Humanoid task, DRDT3 attains performance comparable to the best baseline, CQL, while still delivering a 28.9% improvement over DT. The underperformance observed on certain tasks likely stems from an inherent limitation of sequence modeling methods, which lack dynamic programming capabilities to effectively extrapolate toward higher performance.
>
> Our primary goal, however, is to demonstrate that diffusion models can effectively refine DT-generated actions and enhance DT’s trajectory-stitching capability—an improvement our experiments confirm across a wide range of DT-based baselines and evaluation tasks.
>
> [1] Bhargava, Prajjwal, et al. "When should we prefer decision transformers for offline reinforcement learning?" ICLR, 2024.
>
> ---
>
> *2.	The authors are recommended to provide training curves of the baselines to compare the method performance during the learning phase.*
>
> **Response**:
>
> We appreciate the suggestion to include training curves for baselines to better illustrate performance during the learning phase. We note that while comparing training dynamics across methods would be informative, there are practical challenges in doing so comprehensively:
>
> As our work benchmarks against a wide range of state-of-the-art baselines, including BC, TD3+BC, CQL, IQL, DT, ConDT, QDT, EDT, Synther, Diffuser, Decision Diffuser, AdaptDiffuser, and EDP, reproducing each from scratch (with their unique training setups, hyperparameters, and implementation details) is non-trivial and risks introducing inconsistencies that could bias comparisons. To ensure fairness, we therefore rely on the reported final performance metrics from their original publications.
>
> That said, we agree that tracking learning dynamics is valuable for understanding our method’s behavior. In Figure 3, we provide training curves for DRDT3 and its key variants across representative tasks.
> Moreover, we added training curves comparing convergence behavior across DT, DT3, and DRDT3 on nine Gym tasks as shown in **Figure 4** in our revised manuscript.

---

> ### Author Response · Authors · 2025-08-14
>
> *3.	As an offline RL approach, the stability of the proposed training strategy is suggested to be discussed theoretically.*
>
> **Response**:
>
> We acknowledge the importance of theoretical analysis for understanding the stability of offline RL training strategies, and this is a valuable direction for future work. However, the current focus of our study is on the empirical development and validation of DRDT3 as a practical offline RL framework.
>
> Our work prioritizes designing and evaluating a novel architecture that integrates decision transformers, TTT layers, and diffusion-based refinement to address key empirical challenges in offline RL (e.g., trajectory stitching, handling suboptimal data). The stability of our training strategy is demonstrated through consistent performance across diverse datasets.
>
> While we do not provide formal theoretical guarantees here, we note that the components of our framework build on foundations with existing theoretical insights: Decision Transformer formulates decision making as a sequence modeling problem, TTT layers leverage test-time training for dynamic refinement, diffusion models benefit from convergence properties of Markov chains, and the unified loss draws on principles of multi-task optimization. These connections suggest potential pathways for future theoretical analysis, which we aim to explore to rigorously characterize the stability of DRDT3’s training dynamics.
>
> ---
>
> *4.	The normalized scores of the methods show little improvement on the walker2d-medium-expert-v2 task, as depicted in Figure 3. The paper could benefit from further discussion of this phenomenon.*
>
> **Response**:
>
> Thank you for this valuable feedback. We agree that discussing failure cases enhances the rigor of our analysis. While DRDT3 performs well across most tasks, we acknowledge its underperformance on the Walker2d-Medium-Expert (Walker2d-me) task.
>
> For Walker2d-me, Figure 4 (last subfigure) in our revised manuscript reveals insights into this limitation: we observe a slight degradation in the learning curve as training iterations increase, indicative of overfitting to the dataset’s mixed-quality dynamics. Specifically, as training progresses, the DT3 module memorizes noisy or erratic details from medium-quality trajectories rather than generalizing to the smooth, energy-efficient patterns of expert demonstrations. This leads to increasingly misaligned coarse actions, which the diffusion module cannot fully correct. These might result in degraded policy performance on this task.
>
> We have added this discussion to our revised manuscript.
>
> ---
>
> *5.	Several references require page numbers to meet standard citation practices:*
>
> **Response**:
>
> Thank you for pointing this out. We have carefully reviewed the references and added page numbers where required to comply with standard citation practices.

---

> > ### Comment · Reviewer_7Lmj · 2025-08-20
> > **Official Comments by Reviewer 7Lmj**
> >
> > Thanks for the authors' response.
> >
> > 1. The authors have addressed my previous concern regarding generalization to high-dimensional observations. The additional experiments on more challenging tasks demonstrate the method's applicability to larger state and action spaces.
> > 2. The authors have provided training curves for DRDT3 and its key variants across representative tasks to compare performance during the training phase.
> >
> > However, from the training curves in Figure 4, it can be seen that the performance improvement of the proposed DRDT3 is very slight, especially in the HalfCheetah-me task and the Walker2d-me task. The authors believe the reason is that the DT3 module memorized noisy or erratic details from medium-quality trajectories. While the proposed approach is interesting, its overall improvements remain limited. Therefore, the paper would benefit from further revisions that directly address the shortcomings to achieve more substantial and convincing performance gains.

---

> > > ### Author Response · Authors · 2025-08-20
> > >
> > > We appreciate your careful observation regarding the performance gains of DRDT3.
> > >
> > > We would like to clarify that both DT3 and DRDT3 are our proposed methods, with DRDT3 designed as a further improvement over DT3. We acknowledge that the improvements from DT to DT3 are relatively modest, and it is true that on one task, namely Walker2d-me, DRDT3 exhibits a performance degradation possibly due to noise memorization issues in DT3. **However, the overall performance gains of DRDT3 over DT are substantial, as demonstrated in Tables 3, 4, and 5**. Specifically, DRDT3 achieves average **improvements of 11.9%, 9.4%, 41.9%, 47.8%, and 28.9%** over DT on Gym locomotion, Ant, AntMaze, Adroit, and Humanoid tasks, respectively, which demonstrates that DRDT3 consistently enhances performance across a broad range of benchmarks.
> > >
> > > Regarding Figure 4, the improvements may appear slight because we also included the learning curves of DT3, which lies between DT and DRDT3. If we directly compare DT with DRDT3, the substantial improvements across a wide range of tasks are clearly evident.
> > >
> > > We will revise the manuscript to make this distinction between DT and DRDT3 clearer and to highlight the quantitative improvements more explicitly.
> > >
> > > We hope this addresses your concern regarding the ambiguity of the improvements.

---

### Review · Reviewer_Kk15 · 2025-07-18

**Summary Of Contributions:**

The paper introduces DRDT3, a novel framework for offline RL that improves the DT through two complementary enhancements:
1. DT3 Module: A new sequence modeling component that combines self-attention with Test-Time Training (TTT) layers to better capture temporal dependencies with reduced inference complexity. This module produces coarse action representations.
2. Conditional Diffusion Refinement: A conditional denoising diffusion model is applied to refine the coarse actions predicted by DT3, enabling better stitching of sub-optimal trajectories.
3. Unified Optimization Objective: The paper proposes a joint training objective that integrates both coarse action prediction loss and diffusion refinement loss to ensure consistency between the DT3 module and the diffusion policy.
4. Gated MLP Noise Approximator: A novel noise prediction module is introduced to improve the effectiveness of the diffusion refinement, incorporating both adaptive layer normalization and gated MLPs.

**Audience:**

Yes

**Broader Impact Concerns:**

No significant ethical concerns are raised by this work. The method targets general offline reinforcement learning and could potentially benefit fields such as robotics or healthcare decision-making where safe and efficient offline learning is needed.

**Claims And Evidence:**

Yes

**Requested Changes:**

1. The paper would benefit greatly from enhanced visualization to better communicate the effectiveness of the proposed DRDT3 framework. Currently, the paper includes minimal qualitative or visual analysis, making it difficult for readers to intuitively grasp how DRDT3 improves over DT and other baselines. The authors should include step-by-step visualizations of the action refinement process through the diffusion iterations. Furthermore, adding training curves comparing convergence behavior across DT, DT3, and DRDT3 on representative tasks could further validate the claimed improvements in optimization dynamics.
2. The authors should more explicitly position DRDT3 with respect to Decision Diffuser, Diffuser, and QDT in terms of technical differences and advantages.
3. Add a table or discussion reporting the wall-clock inference time and memory usage of DT3, DT, Diffuser to validate the claim of improved complexity.
4. Include a discussion on failure cases where DRDT3 underperforms or behaves suboptimally (e.g., walker2d-me).
5. Consider releasing code or providing implementation pseudocode for the diffusion refinement module and gated MLP block, to facilitate reproducibility and follow-up research.

**Strengths And Weaknesses:**

Strengths:
1. DRDT3 addresses the key limitation of DT—its poor stitching ability—through a clear and technically justified two-stage approach.
2. Integrating TTT layers into a DT-like model is an effective and original idea for improving sequence modeling under compute constraints.
3. Using diffusion models not as full trajectory generators but as action-level refiners is an elegant way to leverage their generative capacity while maintaining interpretability.
4. DRDT3 consistently outperforms several strong baselines (DT, Diffuser, IQL, QDT) across diverse datasets and settings.
5. The paper carefully evaluates the effect of each component (TTT, diffusion, gated MLP, loss functions), enhancing the credibility of the claims.

Weaknesses:
1. While the combination is new, the core ideas (DT, TTT, diffusion, gated MLP) exist independently and are only moderately novel when integrated.
2. All experiments are on benchmark synthetic tasks; additional results on real-world datasets (e.g., robotics) would enhance practical impact.
3. The paper discusses the reduced complexity of DT3 and fast diffusion refinement, but does not report inference time or memory usage, which is important for applications.

---

> ### Author Response · Authors · 2025-08-14
>
> We sincerely thank you for your insightful comments. We have modified our manuscript accordingly and highlighted all revisions in blue in the updated manuscript.
>
> *1.	While the combination is new, the core ideas (DT, TTT, diffusion, gated MLP) exist independently and are only moderately novel when integrated.*
>
> **Response**:
>
> The innovation of DRDT3 lies not in its individual components but in their synergistic integration, which enables capabilities not achievable by each in isolation:
>
> *Attention–TTT Block*: Uniquely combines self-attention (capturing global dependencies) with TTT layers (dynamic, linear-complexity refinement), yielding superior trajectory modeling compared to Transformer (GPT-2) and SSM (MAMBA) variants.
>
> *Diffusion as Conditional Refinement*: Repurposed from a standalone planner into an iterative refinement module for DT3’s coarse actions, enabling trajectory stitching and overcoming a key limitation of DT-based methods.
>
> *Gated MLP Noise Approximator*: Integrates diffusion timesteps and coarse actions via adaptive normalization, improving noise prediction accuracy.
>
> *Unified Loss*: Jointly optimizes DT3 for coarse action accuracy and diffusion for refinement, ensuring both are aligned with the dataset distribution.
>
> Empirically, this integration delivers substantial improvements over baselines, demonstrating that DRDT3’s novelty extends beyond simply combining existing ideas.
>
> ---
>
> *2.  All experiments are on benchmark synthetic tasks; additional results on real-world datasets (e.g., robotics) would enhance practical impact.*
>
> **Response**:
>
> Thanks for the valuable suggestion. We added experiments to test our DRDT3 on more complex robotics tasks, Adroit-Pen and Door, and a high-dimensional observation task – Humanoid, which has a state-space dimension of 348 and an action-space dimension of 17.
>
> | **Tasks**                    | BC    | CQL   | DT    | DT3 (ours)       | DRDT3 (ours)     |
> |------------------------------|-------|-------|-------|------------------|------------------|
> | pen-human-v1                 | 25.8  | 35.2  | 72.4  | **79.0±2.2**     | 78.3±5.7         |
> | pen-cloned-v1                 | 38.3  | 27.2  | 40.2  | 38.9±20.1        | **57.0±19.8**    |
> | door-human-v1                  | 0.5   | 9.9   | 9.2   | 12.7±6.2         | **19.8±6.3**     |
> | door-cloned-v1                 | -0.1  | 0.4   | 3.7   | 3.7±1.8          | **16.6±5.2**     |
> | **Average**                   | 16.1  | 18.2  | 31.4  | 33.6             | **46.4**         |
> | humanoid-medium               | 13.2  | **49.5** | 23.7  | 29.2±5.8         | 45.9±4.4         |
> | humanoid-medium-expert        | 13.7  | 54.1  | 52.6  | 51.5±2.2         | **62.2±4.2**     |
> | humanoid-expert               | 24.2  | **63.7** | 53.4  | 54.0±2.3         | 59.1±2.0         |
> | **Average**                   | 17.0  | **55.8** | 43.2  | 44.9             | **55.7**         |
>
> **Table 3** in the revised manuscript presents the evaluation performance of DRDT3 on more challenging tasks, including the Pen and Door tasks from the Adroit benchmark and the Humanoid task from Gym. We report results for BC, CQL, and DT as baselines, since other methods are rarely evaluated on these tasks.
>
> As shown, both DT3 and DRDT3 consistently outperform the baseline methods on most Pen and Door datasets, with DRDT3 achieving an average improvement of 47.8% over DT. For the Humanoid task, DRDT3 attains performance comparable to the best baseline, CQL, while still delivering a 28.9% improvement over DT. The underperformance observed on certain tasks likely stems from an inherent limitation of sequence modeling methods, which lack dynamic programming capabilities to effectively extrapolate toward higher performance.
>
> Our primary goal, however, is to demonstrate that diffusion models can effectively refine DT-generated actions and enhance DT’s trajectory-stitching capability—an improvement our experiments confirm across a wide range of DT-based baselines and evaluation tasks.

---

> ### Author Response · Authors · 2025-08-14
>
> *3.	The paper discusses the reduced complexity of DT3 and fast diffusion refinement, but does not report inference time or memory usage, which is important for applications.*
>
> **Response**:
>
> See answer to Q6.
>
> ---
>
> *4.	The paper would benefit greatly from enhanced visualization to better communicate the effectiveness of the proposed DRDT3 framework. Currently, the paper includes minimal qualitative or visual analysis, making it difficult for readers to intuitively grasp how DRDT3 improves over DT and other baselines. The authors should include step-by-step visualizations of the action refinement process through the diffusion iterations. Furthermore, adding training curves comparing convergence behavior across DT, DT3, and DRDT3 on representative tasks could further validate the claimed improvements in optimization dynamics.*
>
> **Response**:
>
> We appreciate the suggestion and have explored step-by-step visualizations of the action refinement process. However, we observed that changes to actions at a single timestep have minimal immediate impact on the subsequent state, even when refined iteratively via the diffusion model (as this refinement still occurs within the same timestep in the trajectory dimension). The effects of refinement become noticeable only over longer horizons—typically beyond 5 timesteps in the trajectory dimension—when the accumulated changes begin to influence the trajectory’s observations.
>
> Moreover, we agree that adding training curves comparing convergence behavior would be beneficial. Therefore, we have included training curves for nine Gym tasks in **Figure 4**. For clarity, each curve is plotted with a moving average (window size = 10) to highlight the trend. As shown, DT3 consistently outperforms DT on most tasks by introducing TTT layers, while DRDT3 further improves performance by refining actions through the diffusion model.
>
> ---
>
> *5.	The authors should more explicitly position DRDT3 with respect to Decision Diffuser, Diffuser, and QDT in terms of technical differences and advantages.*
>
> **Response**:
>
> We appreciate the suggestion and take this opportunity to clarify DRDT3’s positioning relative to Diffuser, Decision Diffuser, and QDT.
>
> *Diffuser* uses diffusion models as a planner in model-based offline RL, requiring first learning environment dynamics and then planning, which can suffer from compounding planner errors.
>
> *Decision Diffuser* samples future states with a diffusion model and infers actions via inverse dynamics. This involves three stages: learning the trajectory distribution, training the inverse dynamics model, and performing state sampling plus action extraction, each introducing potential errors.
>
> *QDT* replaces the return-to-go in Decision Transformer with a value function derived from dynamic programming. This requires learning the value function from offline data, relabeling with its lower bound, and can suffer from value estimation errors.
>
> In contrast, DRDT3 employs the diffusion model solely as an action refinement module, directly operating on the DT output without multi-stage training. It remains a sequence modeling approach without dynamic programming, avoiding the compounding errors of dynamic models and value estimation.
>
> ---
>
> *6.	Add a table or discussion reporting the wall-clock inference time and memory usage of DT3, DT, Diffuser to validate the claim of improved complexity.*
>
> **Response**:
>
> Thank you for the valuable comments. We agree that reporting the wall-clock training and inference time and memory usage is important. Therefore, we added the comparison results in **Table 6** in the revised version. We only compare DT3 with DT since Diffuser introduces the training of a diffusion model, which is not fair to compare with DT3 and DT.
>
> | Model      | Model Size (# param. K) | Hopper-Medium Training (s) | Hopper-Medium Inference (s) | HalfCheetah-Medium Training (s) | HalfCheetah-Medium Inference (s) | Walker2d-Medium Training (s) | Walker2d-Medium Inference (s) |
> |---|-----|------|----|------|--|--|--|
> | **DT**     | 226.8    | 162.2   | **1.7**   | 258.7       | **2.6**     | 239.1     | **2.6**                        |
> | **DT3**    | 234.6     | **134.3**     | 2.6         | **245.8**      | 4.2          | **223.9**                    | 3.1                            |
>
> Table 6 reports the model sizes along with the average training and inference times. As shown, DT3 consistently achieves faster training but fails to surpass DT in inference speed. This is consistent with the findings in the original TTT paper, which reported that TTT layers offer a clear advantage over Transformers during training and become more efficient at inference when processing long contexts. In our case, the sequence length is only 6, a relatively short context, which prevents DT3 from benefiting from TTT’s inference-time speed advantage. However, it still significantly reduces training time and has the potential to save inference time when considering a long context.

---

> ### Author Response · Authors · 2025-08-14
>
> *7.	Include a discussion on failure cases where DRDT3 underperforms or behaves suboptimally (e.g., walker2d-me).*
>
> **Response**:
>
> Thank you for this valuable feedback. We agree that discussing failure cases enhances the rigor of our analysis. While DRDT3 outperforms DT across most tasks (as shown in Table 1), we acknowledge its underperformance on the Walker2d-Medium-Expert (Walker2d-me) task. However, this does not negate its core achievement: enabling effective trajectory stitching via diffusion-based action refinement.
>
> For Walker2d-me, Figure 4 (last subfigure) in our revised manuscript reveals insights into this limitation: we observe a slight degradation in the learning curve as training iterations increase, indicative of overfitting to the dataset’s mixed-quality dynamics. Specifically, as training progresses, the DT3 module memorizes noisy or erratic details from medium-quality trajectories rather than generalizing to the smooth, energy-efficient patterns of expert demonstrations. This leads to increasingly misaligned coarse actions, which the diffusion module cannot fully correct. These might result in degraded policy performance on this task.
>
> We have added this discussion to our revised manuscript.
>
> ---
>
> *8.	Consider releasing code or providing implementation pseudocode for the diffusion refinement module and gated MLP block, to facilitate reproducibility and follow-up research.*
>
> **Response**:
>
> Thank you for the suggestion. We have uploaded our code to an anonymous GitHub repository to facilitate reproducibility: https://anonymous.4open.science/r/DRDT3_Pub-7BB5.
>
> Regarding the pseudocode for the diffusion refinement and gated MLP, we have provided detailed procedures in Algorithm 1 and Algorithm 2, covering both training and inference phases of the refinement process. To make it clearer, we added some comments to assist understanding of the pseudocode in Algorithm 1 and Algorithm 2. Additionally, Figure 1 visually illustrates the refinement workflow, while Figure 2 presents the detailed architecture of the gated MLP block. Since pseudocode may not fully capture the implementation details of the gated MLP, we believe providing the complete code offers a more reliable means for reproducibility.

---

### Decision · Action_Editor_Du91 · 2025-09-09

**Recommendation:** Accept as is

**Additional Comments:**

The paper introduces a novel framework for offline RL by integrating several ideas, including the test-time training (TTT) layers, diffusion, and gated MLP. Though each component exists in the literature, the unification of them for improving decision is novel. The performance is shown to be superior than baselines on a wide range of datasets. The paper include extensive baselines, ablation studies, and sensitivity analyses that clarify the role of diffusion in action refinement and the stitching ability of DT-based methods. I recommend acceptance of this paper and also recommend Reproducibility Certification.

**Audience:**

Yes

**Audience Explanation:**

The method proposed in this paper is interesting to researchers and practioners.

**Claims And Evidence:**

Yes

**Claims Explanation:**

The proposed method is verified on several benchmarks.